# Structural insights into the contactin 1 – neurofascin 155 adhesion complex

Lucas M. P. Chataigner [1], Christos Gogou[2], Maurits A. den Boer [3,4], Cátia P. Frias [2], Dominique M. E. Thies-Weesie[5], Joke C. M. Granneman[1], Albert J. R. Heck [3,4], Dimphna H. Meijer[2] & Bert J. C. Janssen [1] ✉

Cell-surface expressed contactin 1 and neurofascin 155 control wiring of the nervous system and interact across cells to form and maintain paranodal myelin-axon junctions. The molecular mechanism of contactin 1 – neurofascin 155 adhesion complex formation is unresolved. Crystallographic structures of complexed and individual contactin 1 and neurofascin 155 binding regions presented here, provide a rich picture of how competing and complementary interfaces, post-translational glycosylation, splice differences and structural plasticity enable formation of diverse adhesion sites. Structural, biophysical, and cell-clustering analysis reveal how conserved Ig1-2 interfaces form competing heterophilic contactin 1 – neurofascin 155 and homophilic neurofascin 155 complexes whereas contactin 1 forms low-affinity clusters through interfaces on Ig3-6. The structures explain how the heterophilic Ig1-Ig4 horseshoe's in the contactin 1 – neurofascin 155 complex define the 7.4 nm paranodal spacing and how the remaining six domains enable bridging of distinct intercellular distances.

The immunoglobulin-like cell adhesion molecule (IgCAM) family functionally patterns multicellular structures by providing cellular recognition, mechanical support, and contributing to membrane microdomain assembly at adhesion sites between cells[1,2]. Two IgCAM subfamilies, contactin and L1, have roles in the development and function of tissues[3,4], most notably of the nervous system[5,6] by controlling processes of neurite extension, axon guidance, synapse formation, myelination, and axo-glia domain assembly[5–10]. Underlying many of these processes is the formation of oligomers and complexes between and within contactin and L1 family proteins[7,8]. Contactin 1 and neurofascin 155 exemplify the two subclasses and together form a heterophilic adhesion complex at the myelin-axon paranode[9–14] and the neuronal synapse[15]. Dysfunction of contactin and L1 subfamilies are associated with deficits in learning and memory regulation, wide ranging neuropsychiatric diagnoses, neurodevelopmental disorders, cancers, neurodegenerative diseases, and neuropathies[3–6,16–18]. Contactin 1 and neurofascin particularly have well documented roles in the etiology of various demyelinating neuropathies, neurodegenerative diseases including Alzheimer's and Parkinson's, and cancers[3–6,17–19].

Contactin 1 (glycoprotein gp135, F11 in chicken, and F3 in mouse) and five other paralogs (contactin 2–6) form the vertebrate contactin subfamily of the IgCAM family[20]. Defining subfamily features are extracellular six N-terminal immunoglobulin-like domains (Ig), four fibronectin type III domains (FnIII) and posttranslational modifications, i.e., N-glycosylation at multiple sites and a C-terminal glyco-phosphatidylinositol (GPI)-membrane anchor. Crystal structures of one to four domain-containing segments of contactins have shown that the first four domains form a characteristic horseshoe

[1]Structural Biochemistry, Bijvoet Center for Biomolecular Research, Faculty of Science, Utrecht University, Universiteitsweg 99, 3584 CG Utrecht, The Netherlands. [2]Department of Bionanoscience, Kavli Institute of Nanoscience, Faculty of Applied Sciences, Delft University of Technology, Van der Maasweg 9, 2629 HZ Delft, The Netherlands. [3]Biomolecular Mass Spectrometry and Proteomics, Bijvoet Center for Biomolecular Research and Utrecht Institute for Pharmaceutical Sciences, Faculty of Science, Utrecht University, Padualaan 8, 3584 CH Utrecht, The Netherlands. [4]Netherlands Proteomics Center, Padualaan 8, 3584 CH Utrecht, The Netherlands. [5]Van't Hoff Laboratory for Physical and Colloid Chemistry, Debye Institute of Nanomaterials Science, Department of Chemistry, Faculty of Science, Utrecht University, Padualaan 8, 3584 CH Utrecht, The Netherlands. ✉e-mail: b.j.c.janssen@uu.nl

supramodule with versatile binding capacity[21–25], that the Ig5-FnIII2 domains are arranged in an extended head-to-tail conformation and that the FnIII2-3 connection is bent[25]. Membrane anchored or soluble forms of contactins are expressed across tissues with strongest expression in the nervous system[3]. Contactin proteins are critical to neuronal development and homeostasis[16] and function through ubiquitous interactions, which can be homophilic or heterophilic, and *cis* or *trans*[8]. Proteins from the contactin associated proteins (caspr), receptor tyrosine phosphatases (PTPR), amyloid beta precursor protein (APP), notch, and L1 signaling protein families interact with contactins[8,16,26]. The contactin 1−neurofascin 155 *trans* interaction, necessary for proper paranode formation and maintenance, is amongst the best documented interactions with robust cellular level data supporting the concomitant role of these molecules in this setting[27–29].

Neurofascin and three other paralogs (L1, CHL1, and NrCam) form the vertebrate L1 subfamily of the IgCAMs[7] that has a prominent role in nervous system development and homeostasis[6]. All four members are type I transmembrane proteins with six extracellular N-terminal Ig like domains, three to five FnIII domains, various predicted N-glycosylation sites, a single transmembrane spanning helix, and a ~120 cytoplasmic region with a highly conserved ankyrin binding sequence[4]. The only structures available for this subfamily are of neurofascin Ig1–Ig4 that show, similar to the contactins, a characteristic horseshoe supramodule with a conserved homodimerization interface located on Ig1–Ig2[30,31]. Neurofascin consists of over 50 possible splice variants[32], at extracellular and cytosolic sites, with tightly regulated expression patterns intimately associated with their selective biological functions[33,34]. Two isoforms, neurofascin 155 and neurofascin 186, are predominantly expressed in the mature nervous system. Generally referred to as "glial" and "axonal" isoforms respectively, owing to their cell type specific expression, they both play key roles in myelination[11,35,36].

Paranodes flank the nodes of Ranvier and are formed by distal uncompacted loops of myelin that contact the axon at tight 7.4 nm junctions[37]. The contactin 1−neurofascin 155 *trans* complex together with caspr1 are required for the formation and maintenance of these structures, acting as molecular rulers maintaining the rather narrow intercellular spacing[9–14,27–29,35,36]. This ternary complex forms a larger-order assembly that spirals around the axon acting as a membrane diffusion barrier, as denoted by loss of paranodal attachment and axonal compartmentalization observed upon loss of either of the three components[9,10]. Recent research highlights the importance of this barrier as it acts as an incomplete conductive seal for correct action potential propagation[38], and remarkably displays plasticity as the boundary it creates is dynamically displaced over time during learning[39]. Interestingly, contactin 1 and neurofascin 155 also interact at the synapse[15] at a much larger intercellular distance of ~20–25 nm[40–42].

While the contactin 1−neurofascin 155 *trans* complex underlies the attachment of the paranode, many contactin and L1 family members localize and interact in close proximity in nodal, juxtaparanodal, an internodal regions establishing neighboring molecular domains along the myelinated axon[9,10]. How these homologous molecules retain interaction specificities is of great interest in this context, given the high identity between paralogues. It seems part of this inquiry can be answered through the role of isoforms and glycoforms which appear to regulate interaction strength. For contactin 1−neurofascin interactions both N-linked glycosylation of contactin 1[43–45], and splicing of neurofascin[14] are reported to modulate interaction between the two molecules. In the absence of structural information on the contactin 1−neurofascin 155 complex, its interaction mode, adhesion mechanism, parameters determining specificity and role of isoforms and glycosylation have remained unresolved. Moreover, whilst fragments of contactin[21–25] and L1 subfamily[30,31] proteins have informed on the

structural basis of their function, how heterophilic complexes form and their possible stoichiometries has not yet been elucidated.

To address these questions, we structurally characterized the contactin 1 and neurofascin 155 interacting regions, i.e., Ig1–Ig6, individually and in complex. We show that both proteins have a characteristic Ig1–Ig4 horseshoe fold and that Ig5 has distinct connections with Ig4 comparing contactin 1 and neurofascin 155. A conserved surface on Ig1–2 on both molecules plays an important role in complex formation. Several N-linked glycans are involved in intermolecular interactions in the complex and their composition on neurofascin 155 affects interaction affinity. Neurofascin 155 homodimer formation and contactin 1−neurofascin 155 heterocomplex formation are mutually exclusive due to overlapping interaction sites. Structure-based insights are verified by mutations that prevent contactin 1−neurofascin 155 mediated cell clustering. Contactin 1 forms a larger-order zipper in the crystal that is also formed in solution and the neurofascin 155 binding site is exposed in this oligomer. SAXS analysis reveals that the contactin 1 full ectodomain (fe) is elongated and has flexibility. Together our data suggest that the interacting Ig1–Ig4 horseshoes in contactin 1−neurofascin 155 determine the paranodal spacing and that the more extended Ig5-FnIII-4 tails could be used to span the larger intercellular spacing encountered in the neuronal synapse.

## Results

### Contactin 1[Ig1-6]−neurofascin 155[Ig1-6] complex mediated by Ig1–Ig2 domains

Functional analyses have identified the six Ig-domains of contactin 1 and neurofascin 155 as the extracellular regions that mediate trans interactions[14,45,46]. We determined direct interaction of contactin 1[Ig1-6] with neurofascin 155[Ig1-6] with a $K_D$ of 0.22 μM (Fig. 1a, b and Supplementary Fig. 1). Next, we determined the structure of the glycosylated contactin 1[Ig1-6] in complex with neurofascin 155[Ig1-6] from a crystal that diffracted anisotropically to 8–4.8 Å (PDB:7OL4) (Fig. 1a, c–e and Table 1). Structure determination of the low-resolution complex was aided by the structures of the unliganded contactin 1[Ig1-6] (PDB:7OL2) and neurofascin 155[Ig1-6] (PDB:7OK5) that we determined to a resolution of 3.9 and 3.0 Å, respectively. The Ig1–Ig4 horseshoes of contactin 1[Ig1-6] and neurofascin 155[Ig1-6] interact in a near orthogonal edge-on orientation each using a large part of the Ig2 domain side for interaction (Fig. 1c–e, Supplementary Fig. 2, and Supplementary Movie 1). At the center of the heteromeric complex an intermolecular "super" β-sheet is formed by antiparallel hydrogen bonding of the outer G β-strands located in the Ig2 GFC β-sheet of both molecules (Supplementary Fig. 2). At the "bottom side", the super β-sheet forms a concave half barrel that is closed off by the Ig2 CD loops of both molecules and characterized by extensive hydrophobic contacts in the core (Fig. 1e and Supplementary Fig. 2). At the "top side" the intermolecular interactions are less prominent. Here Ig1 AB loops of both molecules contact edges of the opposing Ig2 domains formed by the Ig2 N-terminus and G β-strand (Supplementary Fig. 2). At both sides of the super β-sheet intramolecular salt-bridges are formed. Overall, the contactin 1−neurofascin 155 complex buries ~2089 Å² of solvent accessible area[47] and the interface is of a mixed hydrophobic hydrophilic nature with complementary electrostatic interactions (Fig. 1e).

### Contactin 1−neurofascin 155 interaction mode is conserved in horseshoe-mediated adhesion mechanisms

The Ig1–Ig2 mediated interaction mode for contactin 1 and neurofascin 155 resembles previously reported paralogue contactin 2[22] and L1[22,30] Ig-horseshoe homomeric dimerization modes. This suggests a possible evolutionary relationship between dimerization and heterophilic complex formation, and a likely conserved mechanism for adhesion in these subsets of molecules (Fig. 1e and Supplementary Fig. 2). The contactin 1−neurofascin 155 interaction interface is conserved among vertebrate orthologues indicating that the interface features we

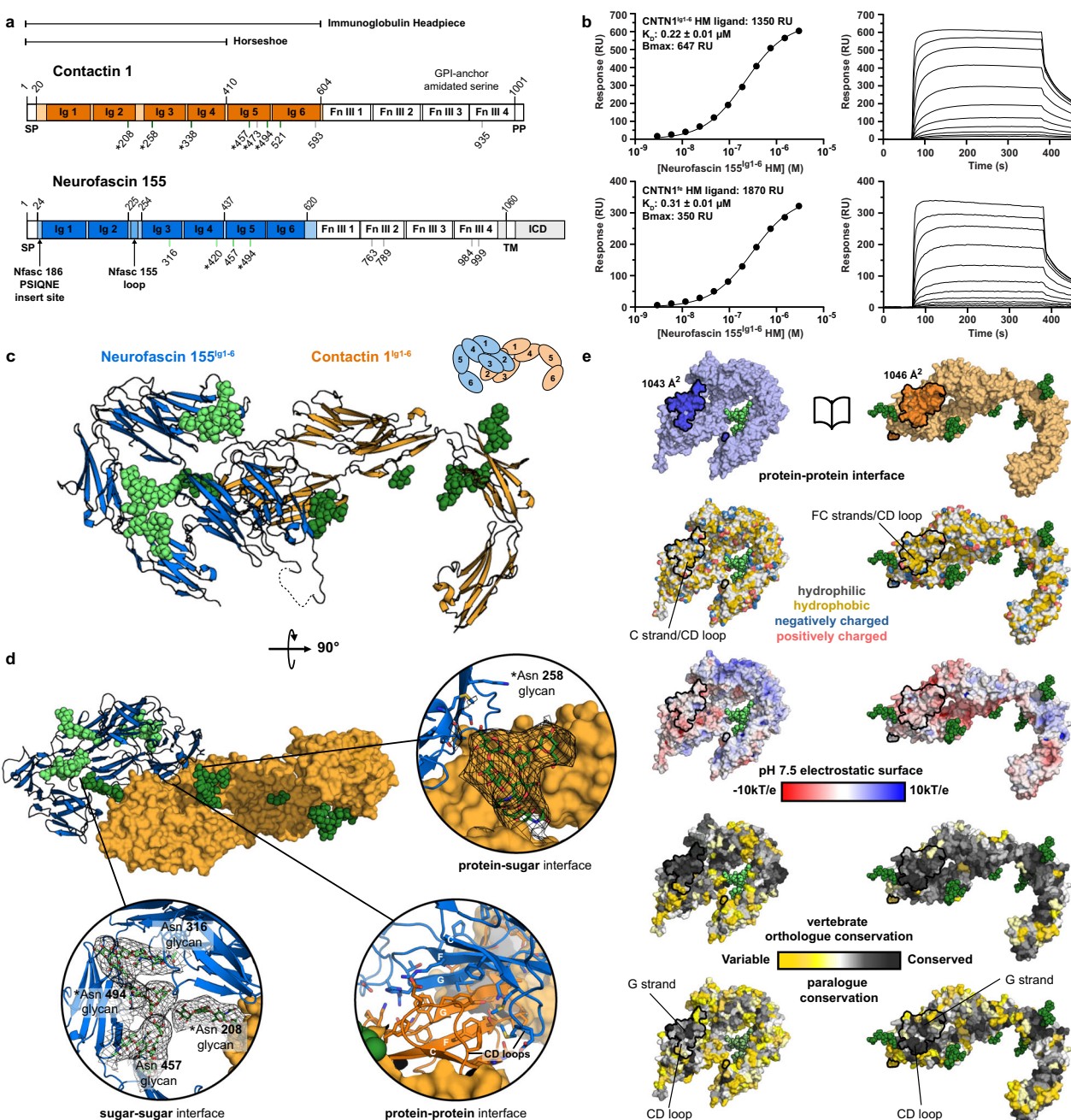

**Fig. 1 | Structural and biophysical insights into contactin 1–neurofascin 155 Ig1–2 mediated complex formation. a** Annotated sequence schematic of contactin 1 and neurofascin 155. Immunoglobulin-like (Ig) and fibronectin type III (FnIII) domains are denoted with domain boundaries (top lines), and crystallized Ig segments colored in. N-glycosylation sites (bottom lines) confirmed by structures (green), predicted (gray), and highly conserved in orthologue conservation analysis (*). Signal peptide (SP), pro peptide (PP), transmembrane domain (TM), glycophosphatidylinositol (GPI) anchor and intracellular domain (ICD) are also indicated. Neurofascin (Nfasc). **b** Surface plasmon resonance imaging interaction data of neurofascin 155$^{Ig1-6}$ containing high mannose glycans (HM) with contactin 1$^{Ig1-6}$ HM (top) or contactin 1$^{fe}$ HM (bottom) ligands. Equilibrium binding data vs. analyte concentration modeled with a 1:1 Langmuir binding model shown left and associated SPR sensorgrams shown right. The affinity of neurofascin 155$^{Ig1-6}$ to contactin

1$^{Ig1-6}$ (top) and to contactin 1$^{fe}$ (bottom) is similar. Contactin 1 (CNTN1), domains Ig1 to Ig6 (Ig1–6), full ectodomain (fe), maximum analyte binding ($B_{max}$), response units (RU). **c** Cartoon representation of contactin 1$^{Ig1-6}$–neurofascin 155$^{Ig1-6}$ complex structure, with glycan residues (spheres), missing loop segment (dashed line) and a schematic indicating the interface. The interface is located on domain Ig1 and 2 of both molecules. **d** Rotated view of the contactin 1$^{Ig1-6}$–neurofascin 155$^{Ig1-6}$ complex structure with contactin 1$^{Ig1-6}$ in surface representation. Insets show various interfaces with glycan density (2mF$_{obs}$-DF$_{calc}$) at 1σ shown as black mesh. **e** Open book representation of the contactin 1$^{Ig1-6}$–neurofascin 155$^{Ig1-6}$ complex in surface representation colored according to various properties and with interaction surfaces outlined. The interface is relatively conserved, suggesting a role in adhesion. Source data are provided as a Source Data file.

identify in the complex structure are likely to be crucial to contactin 1–neurofascin 155 mediated interactions (Fig. 1e). In particular, high conservation of β-strands G and C, CD loop residues and specific N-glycosylation sites observed across orthologues, suggest these features are relevant to heterophilic interaction. To dissect shared features of interaction common to the larger subset of contactin and L1 protein members, we also looked at conservation across paralogues. Plotting conservation of L1 specified paralogues, i.e., L1, CHL1, NrCAM and neurofascin, onto neurofascin 155$^{Ig1-6}$ we found that the G β-strand residues are very conserved, but lower conservation is observed for

**Table 1 | X-ray diffraction data collection and refinement statistics**

| Crystal | CNTN1[Ig1–6] NF155[Ig1–6] | NF155[Ig1–6] | CNTN1[Ig1–6] |
|---|---|---|---|
| **Data collection** | | | |
| Space group | $P2_1$ | $P2_12_12_1$ | $P2_12_12_1$ |
| *Cell dimensions* | | | |
| a,b,c (Å) | 146.8, 151.8, 162.3 | 77.6, 96.3, 238.6 | 92.9, 134.4, 181.7 |
| α,β,γ (°) | 90, 111.8, 90 | 90, 90, 90 | 90, 90, 90 |
| Resolution (Å) | 75.9–4.8 (5.3–4.8)[a] | 79.5–3.0 (3.02–2.97) | 108.1–3.9 (4.3–3.9) |
| $R_{merge}$ | 0.312 (1.926) | 0.168 (1.938) | 0.385 (0.934) |
| $I/\sigma I$ | 4.5 (1.3) | 6.2 (1.0) | 4.4 (1.5) |
| Completeness (%) (spherical) | 48.2 (8.7) | 99.7 (97.4) | 93.9 (77.0) |
| Completeness (%) (ellipsoidal) | 88.5 (73.5) | N/A | N/A |
| Redundancy | 7.1 (7.5) | 6.8 (6.1) | 9.3 (4.1) |
| $CC_{1/2}$ | 0.993 (0.399) | 0.976 (0.399) | 0.961 (0.524) |
| **Anisotropic processing** | | | |
| *Fitted ellipsoid diffraction limit (Å)* | | | |
| 0.999 a* – 0.033 c* | 8.01 | N/A | N/A |
| b* | 4.78 | N/A | N/A |
| −0.298 a* + 0.955c* | 5.35 | N/A | N/A |
| Lowest diffraction limit | 7.85 | N/A | N/A |
| Worst diffraction limit | 8.17 | N/A | N/A |
| Best diffraction limit | 4.8 | N/A | N/A |
| *Overall anisotropy tensor Eigenvectors and Eigenvalues* | | | |
| 0.999 a* – 0.049 c* | 16.94 | N/A | N/A |
| b* | 37.54 | N/A | N/A |
| −0.287 a* + 0.958 c* | 37.44 | N/A | N/A |
| **Refinement** | | | |
| Resolution (Å) | 75.4–4.8 | 60.4–3.0 | 107.9–3.9 |
| No. of reflections | 15662 | 37575 | 20159 |
| $R_{work}/R_{free}$ | 0.29/0.32 | 0.22/0.27 | 0.22/0.27 |
| *No. of atoms* | | | |
| Protein | 18,329 | 9191 | 8996 |
| Ligand/carb | 971 | 355 | 376 |
| Clashscore | 6.1 | 5.6 | 3.9 |
| **B-factors (Å²)** | | | |
| Protein | 257 | 112 | 103 |
| Ligand/carb | 302 | 151 | 148 |
| TLS groups | 24 | 8 | N/A |
| *R.m.s deviations* | | | |
| Bond lengths (Å) | 0.004 | 0.002 | 0.003 |
| Bond angles (°) | 0.79 | 0.55 | 0.54 |
| *Ramachandran (%)* | | | |
| Favored | 93.1 | 97.2 | 95.2 |
| Outliers | 0.0 | 0.1 | 0.0 |
| *Rotamer* | | | |
| Outliers (%) | 3.4 | 1.5 | 0.5 |
| PDB ID | 7OL4 | 7OK5 | 7OL2 |

[a]Values in parentheses are for the highest-resolution shell.
*Denotes reciprocal space.

the C β-strand and CD loop residues (Fig. 1e). For contactin we plotted paralogous conservation with contactin 1 and contactin 2 since initial conservation analyses with the whole family turned up conservation hotspots likely related to function of contactin family binding to receptor tyrosine phosphatases[23–25]. This analysis shows conservation of FC strand and CD loop residues and lower conservation of G strand residues, in a background of overall reduced conservation outside of

the interface area (Fig. 1e). Taken together, the conservation among orthologues and paralogues of residues located in the contactin 1[Ig1–6]–neurofascin 155[Ig1–6] interface suggests that this site may play a role in the adhesion mechanisms within these families.

## Mannose rich glycans on neurofascin 155[Ig1–6] stabilize the interaction with contactin 1

N-glycosylation of contactin 1 has been shown to regulate binding to neurofascin[43–45], with contactin 1 containing mannose rich glycans expected as the relevant form in paranodal adhesion[48]. For structure determination we produced the mannose rich version of the N-linked glycans and we probed both proteins with mannose rich and with complex glycans in binding experiments (Fig. 1b and Supplementary Fig. 1). In the contactin 1[Ig1–6]–neurofascin 155[Ig1–6] complex, mannose rich glycans attached to conserved contactin 1 glycosylation sites at Asn 208 on Ig2, and Asn258 on Ig3, buttress the main protein-protein interface and may form intermolecular glycan-glycan and glycan-protein interfaces, respectively (Fig. 1d). The glycan at contactin 1 Asn208 is in near vicinity to Ig2 of neurofascin 155, while the glycan on contactin Asn258 extends into proximity of glycans attached to Asn457 and Asn494 on Ig5 of neurofascin 155. These additional glycan-mediated interfaces may extend or, more importantly, sterically hinder the protein-protein interface, depending on glycan composition (Fig. 1d and Supplementary Movie 1). We do not observe a strong difference in neurofascin 155[Ig1–6] interaction affinity to contactin 1 immunoglobulin or full ectodomain (fe) segments containing mannose rich glycans (Fig. 1b and Supplementary Fig. 1a–d). Interaction of contactin 1 to neurofascin 155[Ig1–6] containing mannose rich glycans is stronger compared to neurofascin 155[Ig1–6] containing complex glycans although we have not quantified the difference in binding affinity (Supplementary Fig. 1e–g). Possibly the glycans on Ig5 of neurofascin 155, of which Asn494 is conserved, and that interface with the glycan on Asn258 of contactin 1, play a role in modulating the affinity (Fig. 1d).

## Neurofascin 155[Ig1–6] has a distinctive hoe-shaped architecture stabilized by the N-terminus

The conformation of six Ig domains in unliganded neurofascin 155[Ig1–6], determined to a maximum resolution of 3 Å (PDB:7OK5) (Fig. 2a and Table 1), is similar to that of neurofascin 155[Ig1–6] bound to contactin 1[Ig1–6]. The first four domains of neurofascin 155[Ig1–6] display a characteristic horseshoe arrangement in which Ig1 contacts Ig4, and Ig2 contacts Ig3 in an antiparallel fashion enabled by a 180° turn in the Ig2–Ig3 unit connecting loop. As expected, this horseshoe arrangement of Ig1–Ig4 is near identical to that reported for neurofascin 186[Ig1–4] (PDB: 3P3Y)[30] with Cα r.m.s. of 0.83 Å. The 18-residue splice insert present in between domains of the Ig2–Ig3 unit in neurofascin 155, but absent in neurofascin 186, is for a large part unresolved in neurofascin 155 due to flexibility of this extended loop (Figs. 1a and 2a). Domains Ig5 and Ig6 interact head to tail and together make a V-turn with respect to Ig3–Ig4, giving the molecule a distinctive garden hoe like architecture. Here, the horseshoe recalls the tool's blade, and Ig5–Ig6 the handle (Fig. 2a, b and Supplementary Movie 2). At the Ig4–Ig5 tip of the V-turn, neurofascin 155 N-terminal residues 23–31 wedge between these domains and interact intimately with both (Fig. 2a, b and Supplementary Fig. 3a). The Ig4–Ig5 mediated V-turn is stabilized by extensive hydrophobic interactions, in which a central Phe466 of Ig5 is surrounded by Ile25, Ile27, Pro28 and Leu31 of the N-terminus, Ile358 and Leu437 of Ig4, and Val439, Leu444 and Ile 470 of Ig5 (Fig. 2a, b and Supplementary Fig. 3a). In addition, sidechains of Asp30 with Arg442, and Lys355 with Glu493 are poised to form two salt bridges sandwiching the interface. Conservation of the residues involved in the N-terminal, Ig4 and Ig5 interaction suggests the V-turn architecture of Ig4 and Ig5 is a shared feature of L1 family members

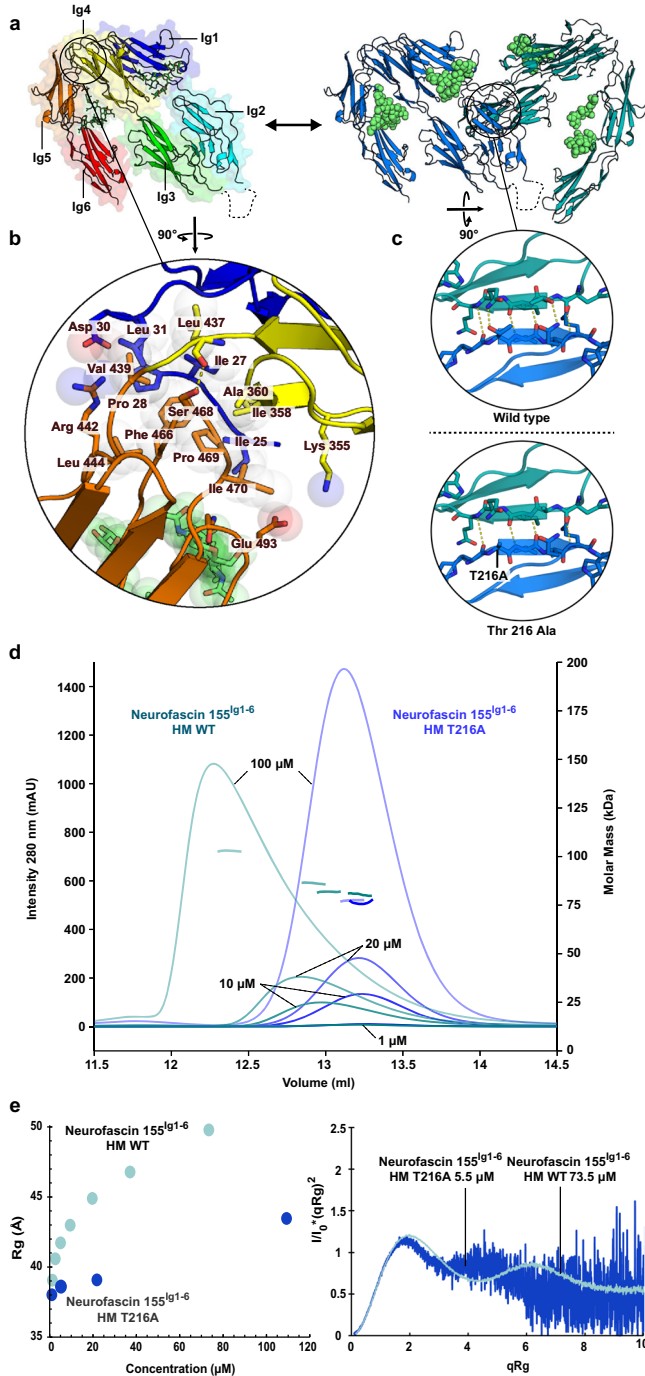

**Fig. 2 | Neurofascin 155$^{Ig1-6}$ has a distinctive garden hoe architecture and dimerizes using an Ig1–2 interface. a** Neurofascin 155$^{Ig1-6}$ monomer (domain rainbow colored) and neurofascin 155$^{Ig1-6}$ dimer (colored by chain) with glycans shown as spheres (lime). The monomer has a hoe-like shape in which the Ig1–Ig4 horseshoe represents the "blade" and domains Ig5 and Ig6, that together make a V-turn with respect to Ig3–Ig4, represent the "handle". Neurofascin 155$^{Ig1-6}$ dimerizes using a similar interface as in the contactin 1$^{Ig1-6}$–neurofascin 155$^{Ig1-6}$ complex. **b** Neurofascin 155 N-terminus and Ig 1, 4 and 5 interface interactions underlying the characteristic architecture. The N-terminus is wedged in between Ig domains 4 and 5. **c** Neurofascin 155$^{Ig1-6}$ dimerization site as wt and as modeled Thr279Ala mutant. **d** Neurofascin 155$^{Ig1-6}$ wt and Thr279Ala mutant SEC-MALS analysis shows the mutant disrupts dimerization (see also Supplementary Table 1). Protein concentrations at injection are indicated. **e** Neurofascin 155$^{Ig1-6}$ wt and Thr279Ala mutant batch SAXS analysis with Rg variations plotted by concentration (left) and characteristic Kratky plots (right). The SAXS data indicate neurofascin 155$^{Ig1-6}$ is a multidomain protein containing flexible regions. The Thr279Ala mutation disrupts neurofascin 155$^{Ig1-6}$ dimerization as determined from the Rg vs. concentration (see also Supplementary Table 2). Domains Ig1 to Ig6 (Ig1–6), full ectodomain (fe), high mannose glycans (HM), complex glycans (CG), wild type (WT), Thr216Ala mutant (T216A), milli absorbance units (mAU). Source data are provided as a Source Data file.

neurofascin 186$^{Ig1-4}$ dimer[30], differing only slightly in conformation of Ig1 EF and AB loops that extend contacts between apposed molecules (Supplementary Fig. 2). At the center of the neurofascin 155 homodimerization interface, established by Ig2 domains with more modest Ig1 contributions, a super β sheet is formed by joining the G β-strands of both Ig2 domains. The hydrophobic interface at the "bottom side" of this super β sheet is smaller in the neurofascin 155$^{Ig1-6}$ homodimer compared to the heterophilic complex, due to a larger hydrophobic area on contactin 1 (Supplementary Fig. 2). The homodimerization interface in neurofascin 155$^{Ig1-6}$ buries ~1700 Å$^2$ of solvent accessible area, slightly less than the 2089 Å$^2$ that is buried in the contactin 1$^{Ig1-6}$–neurofascin 155$^{Ig1-6}$ complex. Neurofascin 155$^{Ig1-6}$ dimerizes in solution as assessed by a concentration-dependent peak shift and molecular weight increase in size-exclusion chromatography with multiangle-light scattering (SEC-MALS), and an increasing particle size at increasing concentration in SAXS analysis (Fig. 2d, e and Supplementary Tables 1 and 2). Consistent with crystallographic observations the type of N-glycosylation does not impact this process (Supplementary Fig. 3d). Neurofascin 155$^{Ig1-6}$ dimerization is substantially reduced by a Thr216Ala mutation reported to disrupt neurofascin 186$^{Ig1-4}$ dimerization[30] by preventing two intermolecular hydrogen bonds in the G β-strands of Ig2 from forming (Fig. 2c). In SEC-MALS analysis the Thr216Ala mutant does not undergo a concentration-depended peak shift nor an increase in molecular weight (Fig. 2d and Supplementary Table 1) indicating dimerization is disrupted. The concentration-dependent increase of the radius of gyration (Rg) in SAXS is reduced by the Thr216Ala mutation, albeit not completely abrogated (Fig. 2e and Supplementary Table 2). The concentration-depended dimerization observed for wild-type neurofascin 155$^{Ig1-6}$ in SEC-MALS and the SAXS Rg increase with mostly monomeric protein below 5 μM, suggests a $K_D$ of dimerization in the ~5–30 μM range (Fig. 2d, e), substantially weaker than the contactin 1$^{Ig1-6}$–neurofascin 155$^{Ig1-6}$ affinity of 0.22 μM (Fig. 1b and Supplementary Fig. 1a, b).

(Supplementary Fig. 3a). Interestingly, L1 and neurofascin 186 insertions within the N-terminus of five and six residues, respectively, may potentially impact the overall architecture (Supplementary Fig. 3a).

### Homodimerization interface and contactin 1 binding site on neurofascin 155 overlap

In the unliganded form, neurofascin 155$^{Ig1-6}$ forms a homodimer (Fig. 2a and Supplementary Movie 2). The dimerization interface overlaps with the binding site for contactin 1, suggesting that competition may exist between neurofascin 155 dimerization and contactin 1 binding (Supplementary Fig. 2). The organization of the neurofascin 155$^{Ig1-6}$ homodimer recalls that of the contactin 1$^{Ig1-6}$–neurofascin 155$^{Ig1-6}$ complex except that a neurofascin 155$^{Ig1-6}$ homodimer molecule is replaced by a contactin 1$^{Ig1-6}$ molecule (Supplementary Fig. 2). In addition, the neurofascin 155$^{Ig1-6}$ dimer is nearly identical to the previously described

### Neurofascin 155$^{Ig1-6}$ has conformational plasticity

The presence of four independent neurofascin 155$^{Ig1-6}$ molecules, two in the contactin 1$^{Ig1-6}$–neurofascin 155$^{Ig1-6}$ crystal and two in the neurofascin 155$^{Ig1-6}$ crystal, allows comparison of conformations (Supplementary Fig. 3b). This reveals that neurofascin 155$^{Ig1-6}$ has conformational plasticity of the Ig2–Ig3 unit and Ig5–Ig6 segments that can be described by two hinging motions. The Ig2–Ig3 unit combination hinges at its connections to Ig1 and Ig4 by a 17° rotation that bends the plane of the Ig1–Ig4 horseshoe. Despite the stabilizing role of the N-terminus, the Ig5–Ig6 combination swings parallel to the Ig1–Ig4

horseshoe by a 19° hinge at the Ig4–Ig5 connection. N-linked glycans play a possible structural role in the structural plasticity of neurofascin 155[Ig1–6]. Glycans at Asn residues 316, 457 and 494 are juxtaposed in the space between the Ig1–Ig4 horseshoe and the Ig5–Ig6 combination (Supplementary Fig. 3c) and may limit the flexion between Ig4–Ig5 depending on the glycan composition. In addition, the glycan on Asn420 of Ig4 interacts with residues on Ig1 and extends the Ig1–Ig4 interface[30] (Supplementary Fig. 3c). The plasticity observed in the neurofascin 155[Ig1–6] crystal structures, is consistent with SAXS measurements for which normalized Kratky plots for both dimer and monomerized neurofascin 155[Ig1–6] have a bimodal and untapered shape expected for a multidomain protein containing flexible regions (Fig. 2d). The structural plasticity does not seem to be associated with either neurofascin 155[Ig1–6] dimerization or with contactin 1[Ig1–6] complex formation as none of the conformations are structurally hindering each other. Most likely the plasticity is an intrinsic property of neurofascin 155 and required for its function as an adhesion protein.

## Contactin 1[Ig1–6] has a distinctive sickle-like architecture that hinges at the Ig4–Ig5 connection

In the structure of unliganded contactin 1[Ig1–6], determined to a maximum resolution of 3.9 Å (PDB:7OL2), the first four Ig domains adopt a typical horseshoe conformation (Fig. 3a). The Ig5–Ig6 combination orients away but nearly in plane with the Ig1–Ig4 horseshoe in a curved architecture giving the molecule a sickle-like shape, with Ig1–Ig4 forming the "handle" and Ig5–Ig6 the curved "blade" (Fig. 3a and Supplementary Movie 3). Overall, the domain orientations of contactin 1[Ig1–6] are remarkably similar between chains within datasets, however comparison of the two unliganded contactin 1[Ig1–6] structures with the two neurofascin 155[Ig1–6] bound reveals a 26° hinge in the Ig4–Ig5 connection (Supplementary Fig. 4b).

Contactin 1 Ig1–Ig4 horseshoe segment resembles previously reported contactin 1, 2, 4 and 5 segments. Our mouse contactin 1[Ig1–6] has Cα r.m.s.d. of 0.64 Å to human contactin 1[Ig2–3] (PDB:3S97)[24], 1.25 Å to chicken contactin 2[Ig1–4] (PDB:1CS6)[21], 1.57 Å to human contactin 2[Ig1–4] (PDB:2OM5)[22], 1.18 Å, 1.63 Å and 2.04 Å to mouse contactin 4[Ig1–4] chains (PDB:3JXA and 3KLD)[23], and 1.14 Å to mouse contactin 5[Ig1–4] (PDB:5E41)[25]. Furthermore r.m.s.d. of individually aligned domains often drops below 1 Å indicating that higher deviations in supramodules are likely due to some conformational freedom of the assembly. Similarly, the Ig5–Ig6 combination of contactin 1[Ig1–6] has Cα r.m.s.d. of 3.58 Å to the homologous combination in mouse contactin 3[Ig5–FnIII2] (PDB:5I99)[25], while individual Ig 5 and 6 domains have 1.10 Å and 0.98 Cα r.m.s.d, respectively.

Stable particle size of contactin 1[Ig1–6] and contactin 1[fe] in the concentration range of 2.5–20 μM measured with SAXS suggests a predominantly monomeric form (Fig. 3b and Supplementary Table 2). However, an upwards curvature of the scattering data at very small angles (Fig. 3c) indicates a small amount of larger species are present. Kratky plots for contactin 1[Ig1–6] and contactin 1[fe] produced with either mannose rich or complex glycans have an untapered shape expected for an extended protein containing flexible regions (Fig. 3d). Similarity of the Kratky plots for mannose rich and complex glycans suggests the overall molecular architecture is not affected by the glycan composition. Contactin 1[fe] Kratky plots have a more pronounced bimodal shape compared to those of contactin 1[Ig1–6] and indicate contactin 1[fe] is more elongated and flexible. Due to the reduced presence of a small amount of larger species, lower concentration SAXS data was used for solution state modeling of contactin immunoglobulin and ectodomain segments. We found contactin 1[Ig1–6] high-mannose SAXS scattering matches well with scattering calculated from the contactin 1[Ig1–6] crystal structure with full mannose trees modeled, with a $\chi^2$ of 0.92 (Fig. 3e, g and Supplementary Table 2). A straightforward model for contactin 1[fe] was generated superposing crystal structure of contactin 1[Ig1–6], with mouse homology models for contactin 1[FnIII1–3], contactin 1[Ig5-FnIII2] and

contactin 1[FnIII4] generated with *Phyre2*[49] based on contactin 1[FnIII1-3] (PDB:5E53)[25] and contactin 3[Ig5-FnIII2] (PDB:5I99)[25]. This model with mannose trees modeled in for confirmed glycosylation sites predicted scattering of the contactin 1[fe] high-mannose SAXS data well, with a $\chi^2$ of 0.82 (Fig. 3f, g and Supplementary Table 2). Overall, the solution state modeling suggests contactin 1[fe] has an extended and serpentine-like architecture.

## A one-dimensional zipper in the crystal packing of Contactin 1[Ig1–6]

Contactin 1[Ig1–6] forms a continuous one-dimensional array along the crystallographic a-axis that is zippered up by Ig3, Ig4, Ig5 and Ig6 mediated interactions, resulting in the c-termini of all molecules orienting on the same side suggesting a *cis*-type interaction (Fig. 4a). The zipper arrangement emerges from four interfaces, denoted α to δ (Fig. 4a), that capture four separate dimeric arrangements of constituting chains (Supplementary Fig. 4a). The dimer burying most surface area buries 2980 Å² and does so using twice the interface α where Ig3–Ig4 on one molecule contact Ig6 of the other molecule (Fig. 4a, b). The second dimeric arrangement involves interface β burying 1190 Å² through Ig3–Ig3 interactions between chain A and a symmetry related chain B (Fig. 4a). A third and fourth arrangement occur through interface γ and δ burying much smaller areas, 420 Å² and 340 Å², respectively (Fig. 4a). In interface γ, Ig3 of chain A contacts Ig5–Ig6 of a symmetry related equivalent molecule, whereas interface δ forms by Ig6–Ig6 interactions between chain B and a symmetry related chain A. Addition of a contactin 1 molecule to a dimer formed through interface α to form a trimer, contributes two additional interaction interfaces, e.g., β and γ (Supplementary Fig. 4a). While a fourth molecule added to a trimer to form a tetramer adds four additional interfaces, i.e., twice α, once γ and once δ (Supplementary Fig. 4a). Several N-linked glycans are poised close to the interfaces. In particular the glycans on Asn258 of Ig3 and Asn521 of Ig6 are located close to intermolecular interaction sites and may influence interactions depending on their composition (Fig. 4a). No steric clashes are apparent upon placing the SAXS-derived contactin 1[fe] model into the zipper organization (Supplementary Fig. 4c), indicating the zipper organization is compatible with ectodomain architecture.

## Contactin 1[Ig1–6] and contactin 1[fe] have a weak propensity to form oligomers

Cell surface interactions may be weak when measured in solution but relevant in a physiological setting due to the stabilizing properties that the membrane attachment provides[50–52]. SAXS analysis of contactin 1[Ig1–6] or contactin 1[fe] in solution, either as a mannose-rich glycan form or as complex glycan form, does not indicate a concentration dependent increase of the Rg up to 10 μM, and only a small increase at 20 μM (Fig. 3b) although the upwards curvature of the scattering data at very small angles at 20 μM (Fig. 3b) may indicate the presence of contactin 1 oligomers or aggregates. We used sedimentation velocity analytical ultracentrifugation (SV-AUC) experiments on the contactin 1 samples at two concentrations to better determine the presence of oligomers (Table 2 and Supplementary Fig. 5). Here c(s) distribution analysis indicates that at low concentration (5 or 9 μM) and at higher concentrations (22 or 36 μM) mannose rich glycosylated contactin 1[fe] and contactin 1[Ig1–6] are primarily monomeric (Supplementary Fig. 5), with some dimer -0.1–11.7%, and trimer species -0.3–1.8% present (Table 2 and Supplementary Fig. 5). In subsets of these samples, higher-order species are present in trace amounts -0.6%. Contactin 1[fe] with complex glycans has slightly higher abundance of dimer, trimer and higher-order species compared to the mannose rich form (Table 2 and Supplementary Fig. 5). Surprisingly, the presence of the dimer and oligomeric species does not seem to be concentration dependent. To test if the oligomers in solution represent the zipper observed in the crystal, we produced a Leu279Arg contactin 1[fe] mutant that we predicted

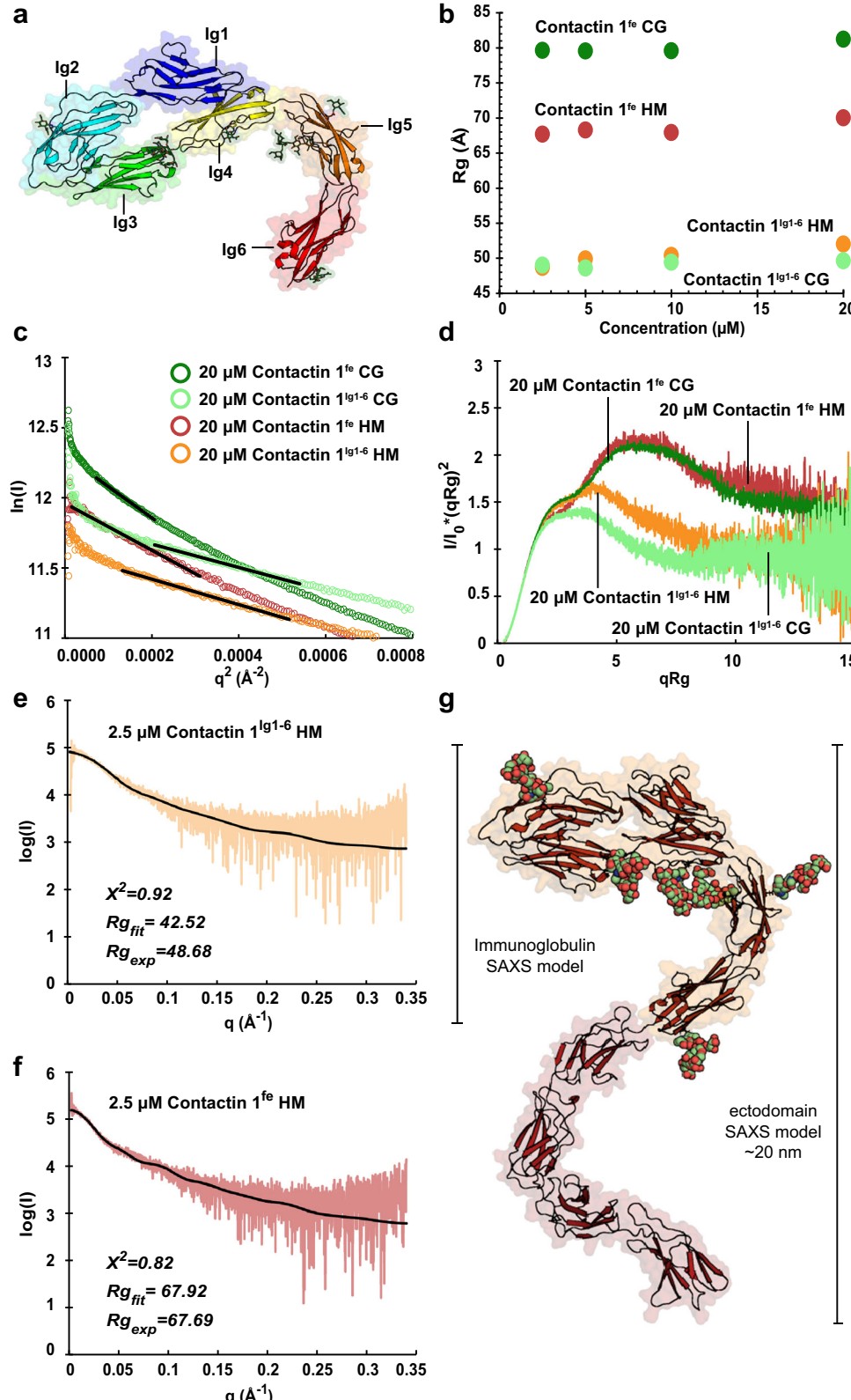

**Fig. 3 | Contactin 1$^{Ig1-6}$ structure has a distinctive sickle architecture and contactin 1$^{fe}$ is more elongated. a** Domain rainbow colored contactin 1$^{Ig1-6}$ monomer. Contactin 1$^{Ig1-6}$ has a sickle-like shape with the Ig1–Ig4 horseshoe forming the "handle" and the Ig5–Ig6 combination forming the curved "blade". **b** Contactin 1$^{Ig1-6}$ and contactin 1$^{fe}$ batch SAXS showing stable Rg across measured concentrations. **c** Batch SAXS guinier plots showing characteristic upturn at low $q$ values indicating the presence of some oligomers. **d** Contactin 1$^{Ig1-6}$ and contactin 1$^{fe}$ characteristic Kratky plots indicating both samples are multidomain proteins containing flexible regions and that contactin 1$^{fe}$ is elongated. **e**–**g** Scattering of HM contactin 1$^{Ig1-6}$ (**e**), and contactin 1$^{fe}$ (**f**). Scattering data calculated for the contactin 1$^{Ig1-6}$ crystal structure with modeled N-glycans (black curve, **e**) and for a contactin 1$^{fe}$ model (**g**) based on superposition of overlapping structures and modeled N-glycans fits well to the experimental scattering data (black curve, **f**). Domains Ig1 to Ig6 (Ig6), full ectodomain (fe), high mannose glycans (HM), complex glycans (CG).

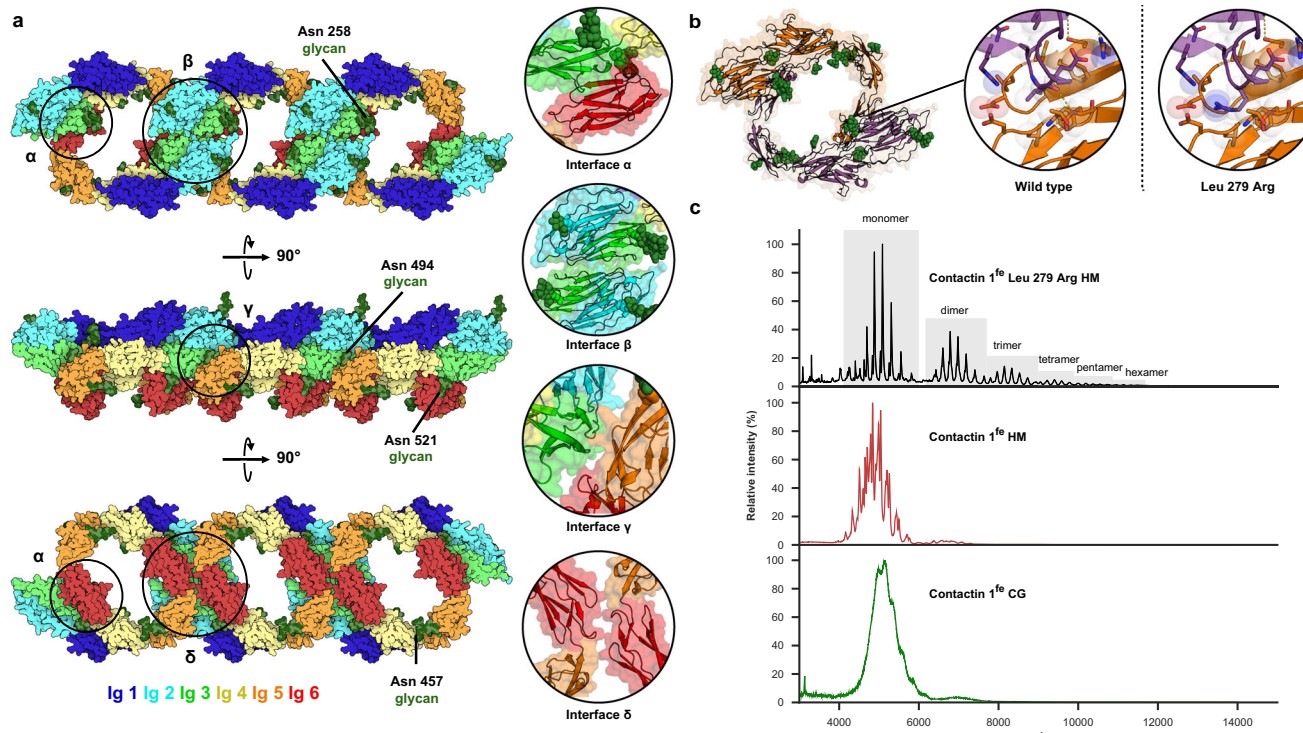

**Fig. 4 | A one-dimensional crystallographic contactin 1$^{Ig1-6}$ zipper and weak propensity of oligomer formation in solution. a** Crystallographic contactin 1$^{Ig1-6}$ zipper with rainbow-colored domains. Interfaces holding molecules together are annotated (α-δ), with insets illustrating these contacts. Glycans in proximity of potential interfaces are annotated. **b** Contactin 1$^{Ig1-6}$ dimer from the zipper burying most surface area through two α interfaces. Inset show details of the interface and a modeled Leu279Arg mutation which increases higher order species abundance. **c** Native mass spectra of contactin 1$^{fe}$ variants reveal a glycan induced heterogeneity and a weak oligomerization propensity for complex glycan (green) and high mannose (red) variants. The propensity to oligomerize is enhanced by the Leu279Arg mutation (black). Full ectodomain (fe), high mannose glycans (HM), complex glycans (CG). Source data are provided as a Source Data file.

**Table 2 | SV-AUC relative abundance of oligomer species**

| Construct | N-Glycosylation | Concentration (μM) | Dimer abundance (%)[a] | Trimer abundance (%)[a] | Trace amounts higher order species[a] |
|---|---|---|---|---|---|
| Contactin 1$^{Ig1-6}$ | HM | 9 | 2.4 | 0.8 | 0.6 |
| Contactin 1$^{Ig1-6}$ | HM | 36 | 0.1 | N/A | no |
| Contactin 1$^{fe}$ | CG | 5 | 9.9 | 3.4 | 4.6 |
| Contactin 1$^{fe}$ | CG | 22 | 9.8 | 4.0 | 4.0 |
| Contactin 1$^{fe}$ | HM | 5 | 11.7 | 1.8 | no |
| Contactin 1$^{fe}$ | HM | 22 | 2.9 | 0.3 | no |
| Contactin 1$^{fe}$ Leu279Arg | HM | 5 | 7.3 | 8.7 | 1.4 |
| Contactin 1$^{fe}$ Leu279Arg | HM | 22 | 5.7 | 6.7 | 3.1 |

[a]Abundances of oligomer species are relative to normalized monomer abundance.

would stabilize the α interface, and thus increase oligomer abundance, by introducing a sterically constrained salt bridge (Fig. 4b). Indeed, the Leu279Arg contactin 1$^{fe}$ produced with mannose rich glycans has increased trimer ~6.7–8.7% and higher-order species, compared to wt protein in SV-AUC analysis (Table 2 and Supplementary Fig. 5).

We used native mass spectrometry (MS) to corroborate the oligomerization patterns observed in SV-AUC and to determine accurate masses for the different constructs (Fig. 4c). Native MS measures masses of intact proteins and their complexes under native-like conditions, allowing noncovalent interactions to remain intact[53]. As expected, native mass spectra of contactin 1$^{fe}$ harboring complex glycans (contactin 1$^{fe}$ CG), revealed that this construct is highly heterogeneous (Fig. 4c). From this data we can only partly resolve a charge-state distribution (4000 < $m/z$ < 6000) and used that to estimate a mass of around 129 kDa. The spectrum also indicates the presence of a low abundant population for a dimer (6500 < $m/z$ < 7500). The glycosylation micro-heterogeneity becomes greatly reduced in mannose-rich produced material that contains less complex glycans (contactin 1$^{fe}$ HM). From the charge-resolved data we obtain a more accurate mass of around 121 kDa, matching with the smaller and more homogeneous glycans, and observe again also low abundant dimeric species. The Leu279Arg mutant of contactin 1$^{fe}$ produced with mannose rich glycans showed the best resolved spectra and strongest oligomerization propensity. In these spectra, next to the 122 kDa monomer, charge state distributions for oligomers up to hexamers could be resolved, with the abundance decreasing with oligomer size. Taken together, the data from SV-AUC and native MS indicate that contactin 1$^{fe}$ has a weak propensity to oligomerize, which can be enhanced by the Leu279Arg mutation.

## Cell-clustering assays to substantiate the structural insights
Seeking further validation of structural insights into contactin 1 and neurofascin 155 adhesion, we set up a previously reported cell clustering assay to characterize adhesion complex formation[21,54,55].

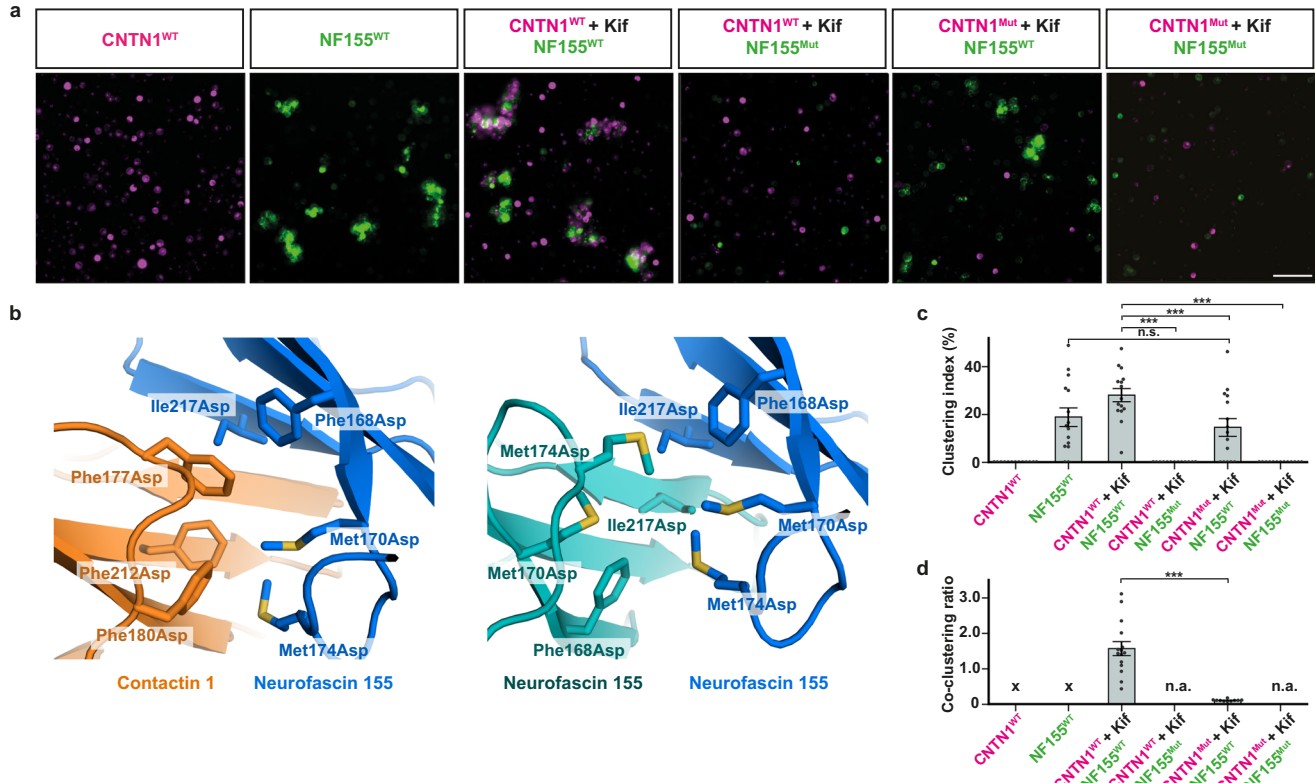

**Fig. 5 | Contactin 1–neurofascin 155 expression mediates cell co-clustering.**
**a** Representative cell clustering images of K562 cells expressing contactin 1 (mCherry; magenta; CNTN1) and neurofascin 155 (GFP; green; NF155). Wildtype contactin 1–neurofascin 155 co-clusters form when contactin 1 is expressed in the presence of kifunensine (Kif; 10 μM). Neurofascin 155 but not contactin 1-expressing cells exhibit homophilic clustering. Mutation of the competitive interface residues on either protein abolishes co-clustering. Each experiment was repeated three times independently with similar results. Scale bar, 100 μm. Contactin 1 Phe177Asp, Phe180Asp and Phe212Asp is CNTN1$^{Mut}$, neurofascin 155 Phe168Asp, Met170Asp, Met174Asp and Ile217Asp is NF155$^{Mut}$. **b** Location of the mutations. Contactin 1–neurofascin 155 interface residues, shown in stick representation, at the "bottom side" of the Ig2 super β-sheet (left panel) are mutated to aspartates and prevent heterophilic *trans* interactions. The same neurofascin 155 residues are also located in the neurofascin 155 homodimer interface (right panel) and mutating them to aspartates prevents homophilic *trans* interactions. **c** Clustering index; the proportion of the total segmented cell area classified as clusters. *p* values are: 0.9544

for NF155 vs. CNTN1mut + Kif/NF155, <0.0001 for CNTN1 + Kif/NF155 vs. CNTN1 + Kif/NF155mut, 0.0003 for CNTN1 + Kif/NF155 vs. CNTN1mut + Kif/NF155 and <0.0001 for CNTN1 + Kif/NF155 vs. CNTN1mut + Kif/NF155mut. **d** Co-clustering ratio determined as the mean mCherry signal over the mean GFP signal per cluster. The increase of this ratio for contactin 1 (kifunensine-treated wildtype)–neurofascin 155 (wildtype) indicates co-clustering, whereas the low ratio values for the other conditions indicate presence of only neurofascin 155 homophilic clusters. Single data points in **c** and **d** represent the average values from the *n* = 15 (*n* = 14 for contactin 1$^{Mut}$–neurofascin 155$^{Mut}$) images from *N* = 3 independent experiments (each experiment 5 images). Error bars indicate the mean ± SEM. Statistical significance was determined by performing a one-way ANOVA followed by a Tukey's multiple comparison test, and results are indicated using the following conventions: n.s. not significant, ***$p$ < 0.001. In absence of clusters for analysis "n.a." is indicated, and "X" denote single-channel conditions in which no fluorescence ratio could be calculated. *p* value is <0.0001 for CNTN1 + Kif/NF155 vs. CNTN1mut + Kif/NF155. Source data are provided as a Source Data file.

Previous reports have suggested, that in the cellular context a high-mannose form of contactin 1 is a pre-requisite for neurofascin 155 binding[44]. We confirmed this finding by treating contactin 1-expressing K562 cells with the mannosidase I inhibitor kifunensine, resulting in high-mannose type glycosylation. Contactin 1-expressing cells did not cluster by themselves, irrespective of kifunensine treatment (Fig. 5 and Supplementary Fig. 6). This shows that contactin 1 does not engage in homophilic *trans* interactions, consistent with our finding that the interfaces we observe in contactin 1 oligomers most likely represent *cis* interactions. On the other hand, neurofascin 155-expressing cells engaged in homophilic cell-clustering, supporting previous reports of neurofascin mediated homophilic adhesion[30], and this interaction was not influenced by kifunensine treatment (Fig. 5 and Supplementary Fig. 6). Kifunensine-treated contactin 1-expressing cells co-clustered with neurofascin 155-expressing cells as indicated by the co-clustering ratio that shows the clusters contained both contactin 1 and neurofascin 155-expressing cells (Fig. 5d). Untreated cells did not form such heterophilic co-clusters, i.e., the clusters consisted predominantly of neurofascin 155-expressing cells (Supplementary Fig. 6c, d). Overall,

we confirm previous reports[44] that presence of high mannose glycans on contactin 1 are required for heterophilic interactions with neurofascin 155 in the cellular setting. Interestingly, we additionally found that contactin 1-expressing cells not treated with kifunensine and thus containing complex glycans, co-clustered with neurofascin 155-expressing cells as long as those cells were treated with kifunensine (Supplementary Fig. 6d). When both cell types were treated with kifunensine, i.e., contactin 1 and neurofascin 155 both have high-mannose glycans, no co-clustering occurred (Supplementary Fig. 6d). These findings suggest that specific glycosylation types and patterns are required for interaction of contactin 1–neurofascin 155 in the cellular context.

Based on the contactin 1–neurofascin 155 structure we designed two interface mutant versions in which hydrophobic residues at the "bottom side" of the Ig2 GFC super β-sheet (Fig. 1d and Supplementary Fig. 2d) are changed to charged residues that we predict would prevent contactin 1–neurofascin 155 heterophilic and neurofascin 155 homophilic *trans* interaction. Surface residues Phe177, Phe180 and Phe212, were mutated to aspartate in contactin 1$^{Mut}$ and surface residues

Phe168, Met170, Met174 and Ile217 were mutated to aspartate in neurofascin 155[Mut] (Fig. 5b). As expected, cells expressing contactin 1[Mut] (treated with kifunensine) and cells expressing neurofascin 155[Mut] did not co-cluster (Fig. 5a, c, d). In addition, contactin 1[Mut]-expressing cells (treated with kifunensine) did not co-cluster with neurofascin 155[WT]-expressing cells, nor did neurofascin 155[Mut]-expressing cells co-cluster with contactin 1[WT]-expressing cells (treated with kifunensine) (Fig. 5a, c, d). These experiments illustrate that the mutant versions independently abolish co-clustering and indicate that the hydrophobic surfaces we identify, both on the contactin 1 and neurofascin 155 Ig2 domains, are required for heterophilic *trans* interaction. Furthermore, as expected, neurofascin 155[Mut]-expressing cells did not form homophilic cell clusters either (Fig. 5 and Supplementary Fig. 6), confirming that the heterophilic adhesion site of contactin 1–neurofascin 155 overlaps with the homophilic adhesion site of neurofascin 155.

Taken together, the cell clustering experiments show that contactin 1–neurofascin 155 *trans* interaction is mediated through Ig2 as observed in the complex structure, that neurofascin 155 homophilic *trans* interaction is mediated through the interface we observe in the neurofascin 155[Ig1−6] dimer structure and, as expected, that contactin 1 does not engage in homophilic *trans* interactions.

## Discussion

Members of the contactin and L1 families act concomitantly through functionally diverse oligomers and complexes to establish adhesion underlying neuronal tissue wiring and connectivity[7,8]. The consensus view from structural and biophysical methods has been that in these families the first Ig1–4 domains have a characteristic backfolded horseshoe architecture[21–25,30,31,46,57]. In many cases, immunoglobulin domains and particularly the horseshoe supramodule, have been established as necessary and sufficient for trans adhesion[14,21,22,30,31,45,46,57–63]. Here we show that the mouse contactin 1[Ig1−6]–neurofascin 155[Ig1−6] complex, and the neurofascin 155[Ig1−6] dimer are formed via Ig1–2 interactions between the horseshoe modules in a mode similar to that described for the human contactin 2[Ig1−4] [22] and neurofascin 186[Ig1−4] [30] horseshoe homodimers. This common mode of interaction suggests that the heterophilic interaction between contactin 1 and neurofascin 155 may have emerged from duplication of ancestral homodimerizing proteins[7,8]. Heterophilic interactions emerging in protein families from ancestral homodimers are proposed to lead to heterodimeric species with varying affinities that may be exploited to finetune spatio-temporal control of cellular responses[64]. This notion is supported on a subcellular level by the similarity in interaction mechanisms for the contactin 1–neurofascin 155 heterophilic *trans* complex that functions in myelin paranode junction formation[11–13] and the contactin 2 homophilic *trans* complex that is required for myelin juxtaparanode formation[9,10] (Fig. 6).

Contactin 1 and neurofascin 155, together with Caspr 1, control paranodal adhesion[11–13,48] and maintain the intercellular distance of 7.4 nm ± 0.6 nm[37] important for maturation and homeostasis of myelinated fibers[27–29] ensuring effective saltatory conduction[38,39]. The contactin 1[Ig1−6]–neurofascin 155[Ig1−6] complex has an edge-on size of ~7 nm (Fig. 6) indicating that both proteins must lay flat between the two cellular surfaces with the horseshoe modules determining the paranodal spacing, similar to what has been proposed for horseshoe-containing sidekick proteins in the retina[65]. Interestingly, contactin 1 and neurofascin 155 are also found[66,67] and interact in the neuronal synapse[15] which has a much larger intercellular distance of ~20–25 nm[40–42]. The size of contactin 1[fe], of 20 nm as determined from SAXS (Fig. 3), together with the dimensions of the neurofascin 155 ectodomain, plasticity in the Ig4–5 connection that we observe for both contactin 1 and neurofascin 155 (Supplementary Figs. 3 and 4), flexibility reported by others in the FnIII 1–3 contactin segment[25], and flexibility at the membrane attachment sites may allow to span the intercellular synapse space (Fig. 6) and enable these molecules to

function at different cell-cell distances. If and how structural plasticity of contactin 1 and neurofascin 155 plays a role in cellular adhesion will require further functional verification. The expression levels of specific isoforms[11,33,34], the localization in space and time[7,8] and post-translational modification of contactin 1[26–28] and neurofascin[14,68–70] impact on the development and function of the nervous system. Neurofascin 155 homodimerization and contactin 1–neurofascin 155 complex interfaces overlap, indicating these interactions exclude one another. The contactin 1–neurofascin 155 complex has a higher affinity (0.22 μM, Fig. 1b) compared to neurofascin 155 dimerization (5–30 μM), suggesting a preference for heterophilic complex formation over homodimerization. This preference also explains why the neurofascin 155 homophilic interaction does not prevent contactin 1–neurofascin 155 heterophilic interaction in the cell clustering assay (Fig. 5 and Supplementary Fig. 6). In the biological context however, factors affecting effective concentration of components, such as expression levels, anchoring and trafficking processes, distances between molecules on apposed membranes, and whether the competing interactions can occur in a *cis* or in a *trans* setting, likely provide additional modulation of this preference. In addition, neurofascin has many isoforms[33] and their expression levels and location are often distinct[11,33–36,71,72]. Comparing the structures of neurofascin 155[Ig1−6] and neurofascin 186[Ig1−4] shows that the contactin 1 binding site on both molecules is very similar. Given isoform splice differences are outside of the Ig1–Ig2 region, all neurofascin isoforms could likely bind contactin 1 using the interface reported here on Ig1–Ig2. In line with our model, contactin 1 would be able to interact with any neurofascin isoform, a finding supported by earlier neurofascin isoform–contactin 1 interaction studies that show unambiguous binding with differing relative efficiencies[14]. The distinctive neurofascin 155[Ig1−6] hoe-shaped architecture may be regulated by splicing at the N-terminus (Fig. 2a, b and Supplementary Fig 3), possibly affecting the presentation of the contactin 1 binding site and explaining splice-site dependent difference in interaction[14] and absence of interaction in a cellular context upon deletion of the Ig5–6 domains in neurofascin[46]. Additionally, N-linked glycosylation has been shown to regulate binding of contactin 1 and neurofascin 155 in a cellular context[43–45] and we show in this context that high-mannose glycans, either on contactin 1 or on neurofascin 155 but not on both, enable heterophilic cell-cell interactions. In SPR experiments we show that contactin 1–neurofascin 155 interactions still occur when both proteins have the same high-mannose glycan type (Fig. 1b), suggesting that glycan microheterogeneity, steric properties of the full-length molecules, or structural constrains of the cellular context may additionally modulate transcellular interactions. From the contactin 1[Ig1−6]–neurofascin 155[Ig1−6] structure, conserved glycosylation sites in the immunoglobulin segments appear poised to extend or sterically hinder interaction depending on glycosylation microheterogeneity. Taken together, a balance of pre- and post-translational modification, localization and protein concentrations underlies contactin 1–neurofascin interaction.

While contactin 1 and neurofascin 155 interact in a mode similar as to neurofascin[30] (Supplementary Fig. 2) and contactin 2[22] homodimerization, in the structure we report contactin 1 does not form homodimers via the Ig1–Ig2 interface. Instead, contactin 1[Ig1−6] uses domains Ig3, Ig4, Ig5 and Ig6 to form a larger-order one-dimensional zipper that leaves the neurofascin 155 binding site accessible. While we cannot exclude that under specific conditions contactin 1 Ig1–Ig2 interface interactions may occur, contactin 1 does not appear to form homo-*trans* interactions as indicated by our cell-clustering assays and by others[73]. Contactin 1 may however form *cis* interactions, and the zipper is compatible with *cis*-interactions as the c-termini, that connect to the cell surface, are all on the same side. While interactions measured in solution can be weak, they may be relevant in a physiological membrane-associated setting that can provide additional stability[50–52,74–76] and contactin 1 expression levels are particularly high

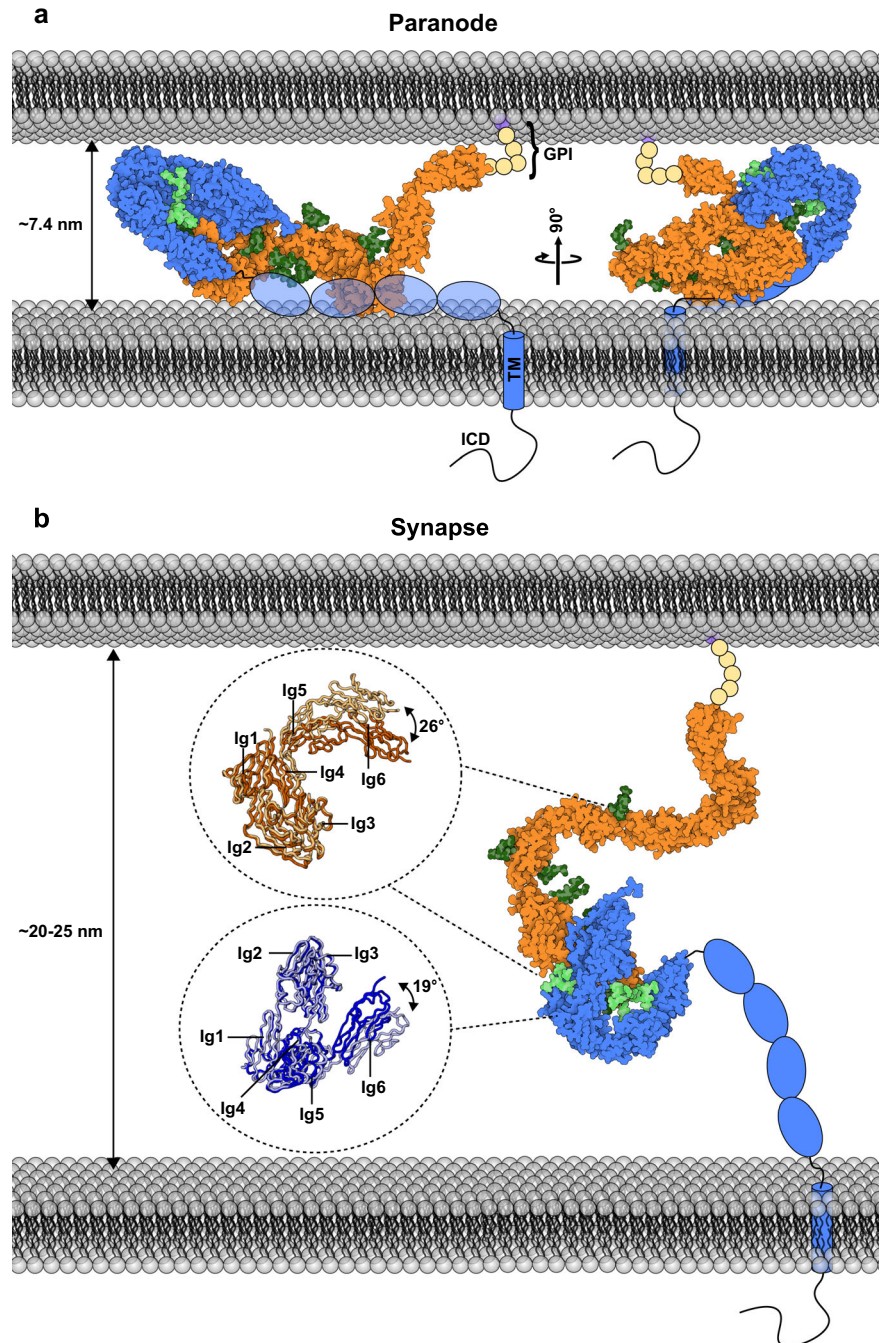

**Fig. 6 | Contactin 1–neurofascin trans adhesion can span distinct intercellular distances. a** Model of the contactin 1–neurofascin 155 complex in the ~7.4 nm paranodal intermembrane spacing, with molecules laying flat on the membranes and horseshoes determining minimum intermembrane distance. **b** Model of the contactin 1–neurofascin complex spanning the ~20–25 nm synaptic cleft, enabled by the length and flexibility of the C-terminal domains. The models are based on the crystal structures and the SAXS data with missing FnIII domains of neurofascin shown as schematic ovals. Glycophosphatidylinositol (GPI) anchor, intracellular domain (ICD).

in axonal membranes[77]. Interestingly, linear arrays of contactin 1 and caspr or neurofascin 155 and caspr have been reported in the paranode and along the axon[48,78,79] and a ternary complex of contactin 1–neurofascin 155–caspr1 is believed to form axon-encircling septate-like junctions in the paranode[11–13,48]. It is currently not clear if the zipper we identified plays a role in this setting and although the neurofascin 155 binding site is available in the zipper, not all sites can be occupied by neurofascin 155 as this would lead to steric clashes between neurofascin 155 molecules. Finally, secreted contactin 1 plays a role in mediating nodal sodium channel clustering[80] and it is possible that the oligomeric form of contactin 1 has a role in this function, but this has not been experimentally verified.

Evidence in recent years has emerged implicating both contactin 1 and neurofascin 155 as central players in a host of pathologies. Both proteins are intimately associated with autoimmune neuropathies[81], such as multiple sclerosis[19,82]. Furthermore, contactin may play a role in neurodegenerative diseases with reports implicating it in Parkinson's[83] and Alzheimer's disease[84,85], but also cancers where it impacts cancer progression and metastasis[86,87]. Our work on the contactin 1–neurofascin 155 complex and the individual proteins, provides a

steppingstone for the development of future therapeutics targeting these molecules.

## Methods

### Construct generation and mutagenesis

Contactin 1 (CNTN1) based on Image Clone 30099512, and human cell line expression codon optimized neurofascin 155 (NF155) based on NCBI transcript variant NM_001160316.1 obtained from (GeneArt Thermo Fisher), were used as templates to generate contactin 1[Ig1–6] (residues 21–604), neurofascin 155[Ig1–6] (residues 25–633), contactin 1[fe] (residues 21–996) and neurofascin 155[fe] (residues 25–1059) constructs by polymerase chain reaction (PCR) (Supplementary Tables 3 and 4). Neurofascin 155[Ig1–6] Thr216Ala, contactin 1[fe] Leu279Arg, contactin 1[fe] Phe177Asp, Phe180Asp, Phe212Asp (contactin 1[Mut]) and neurofascin 155[fe] Phe168Asp, Met170Asp, Met174Asp, Ile217Asp (neurofascin 155[Mut]) mutants were created using overlapping primers (Supplementary Table 3). All constructs were subcloned using BamHI/NotI sites into pUPE107.03 (cystatin secretion signal peptide, C-terminal His6 tag, U-Protein Express), except for contactin 1[Mut] that was subcloned into pUPE107.58 (cystatin secretion signal peptide, C-terminal transmembrane helix–mCherry fluorophore–His6 tag, U-Protein Express) and for neurofascin 155[Mut] that was subcloned into pUPE107.21 (cystatin secretion signal peptide, C-terminal transmembrane helix–eGFP fluorophore–His6 tag, U-Protein Express). Contactin 1[Ig1–6] and contactin 1[fe] were also further subcloned into pUPE107.62 (cystatin secretion signal peptide, C-terminal biotin acceptor peptide-His6 tag) vector (U-Protein Express). Contactin 1[fe] and neurofascin 155[fe] were also subcloned into pUPE107.58 and pUPE107.21, respectively (called contactin 1[WT] and neurofascin 155[WT]).

### Protein expression and purification

Complex glycan (CG) proteins were produced in suspension preparations of Epstein–Barr virus nuclear antigen I (EBNA1)-expressing HEK293 cells (HEK293-E) (U-Protein Express), while high mannose (HM) proteins were produced in N-acetylglucoaminyltransferase I-deficient (GnTI−) EBNA1-expressing HEK293 cells (HEK293-ES) (U-Protein Express). Medium was harvested 6 days after transfection and cells were spun down by 10 min of centrifugation at $1000 \times g$. Cellular debris was then spun down from medium for 15 min at $4000 \times g$. Protein was purified using Ni Sepharose excel (GE Healthcare) affinity chromatography followed by size exclusion chromatography (SEC) on either Superdex200 Hiload 16/60 (GE Healthcare) or Superdex200 10/300 (GE Healthcare) columns equilibrated in SEC buffer (25 mM HEPES pH 7.5, 150 mM NaCl). For Leu279Arg contactin 1 samples cOmplete™, Mini, EDTA-free protease inhibitor cocktail was additionally added to SEC buffer. Protein was then concentrated to 5–10 mg ml⁻¹ and stored at −80 °C. Purity was evaluated by SDS-PAGE and Coomassie staining.

### Crystallization and X-ray data collection

Sitting-drop vapor diffusion at 4 °C was used for all crystallization trials, by mixing 150 nl of protein solution with 150 nl of reservoir solution. Crystals of contactin 1[Ig1–6]–neurofascin 155[Ig1–6] (equimolar ratio final mixture 6 mg ml⁻¹) grew from a condition with 2% v/v Tacsimate™ pH 5.0, 0.1 M Sodium citrate tribasic dihydrate pH 5.6, and 16% w/v PEG 3,350. Crystals of neurofascin 155[Ig1–6] (8 mg ml⁻¹) grew from a condition with 20% (w/v) PEG 8000 and 100 mM HEPES/ Sodium hydroxide pH 7.5. Crystals of contactin 1[Ig1–6] grew from a condition set up with contactin 1[Ig1–6]–neurofascin 155[Ig1–6] (equimolar ratio final mixture 6 mg ml⁻¹) with 20% w/v PEG 3350 and 0.2 M Magnesium nitrate hexahydrate pH 5.9. Reservoir solution supplemented with 30% of glycerol was added as cryo-protectant to the crystals before plunge freezing them in liquid nitrogen. Data sets were collected at 100 K at Diamond Light Source beamline I03 (contactin 1[Ig1–6]–neurofascin 155[Ig1–6] and neurofascin 155[Ig1–6], at a wavelength of 0.9763 Å), and Diamond Light Source beamline I24 (contactin 1[Ig1–6], at a

wavelength of 0.9686 Å). The contactin 1[Ig1–6] dataset was collected using helical collection strategy.

### Structure determination and refinement

Integrated data were obtained for neurofascin 155[Ig1–6] and contactin 1[Ig1–6] datasets, from the xia2 dials diamond beamline data auto processing pipeline[88], and further processed in *AIMLESS*[89]. Resolution limit cut off was determined based on mean intensity correlation coefficient of half-data sets, CC1/2. Unmerged and unscaled data were obtained from the xia2 dials diamond beamline data auto processing pipeline[88] for two isomorphic contactin 1[Ig1–6]–neurofascin 155[Ig1–6] datasets collected in succession from the same crystal. These data were used to produce a combined unmerged and unscaled dataset using *POINTLESS*[90]. Anisotropic cut-off, merging and scaling of the combined dataset was performed by the *STARANISO*[91] webserver. Following recommendations from the *STARANISO*[91] webserver, unobserved and unobservable reflections that lie outside the diffraction cut-off surface were removed from the merged data file. All structures were solved by molecular replacement using *PHASER*[92]. Initial search models for unliganded neurofascin 155[Ig1–6] and contactin 1[Ig1–6] structures were (PDB: 3P3Y)[30] for neurofascin Ig1–4 residues 31–437, (PDB: 2OM5)[21] for contactin Ig1–4 residues 38–410, and *PHYRE2*[49] models generated with >90% confidence for Ig5 and Ig6 domains of both neurofascin 155 and contactin 1. For contactin 1[Ig1–6]–neurofascin 155[Ig1–6] complex structure, refined structures from unliganded datasets were used as search models. Structure refinement of unliganded datasets was performed using *PHENIX*[93] with automatic weighting options. Manual model building was done in *COOT*[94]. Manual inspection and correction of unliganded structures was performed iteratively in *COOT*[94] in between automated refinement runs to correct register errors. Readjusting the positioning of residues or correcting the register was done on the following parts: neurofascin 155 Ig 5 residues 445–449 and 482–492, neurofascin 155 Ig 6 residues 564–584, contactin 1 Ig 5 residues 415–422, and contactin 1 Ig 6 residues 549–566. To minimize overfitting of the unliganded neurofascin 155[Ig1–6], and unliganded contactin 1[Ig1–6] models, secondary structure and NCS restraints were applied. Unliganded neurofascin 155[Ig1–6] was additionally refined with TLS parameters given the observed mobility of domains between chains within the dataset. Structure refinement of the contactin 1[Ig1–6]–neurofascin 155[Ig1–6] dataset was initially performed using *REFMAC*[95] with manual model building performed in *COOT*[94]. Given the low resolution and highly anisotropic data for the contactin 1[Ig1–6]–neurofascin 155[Ig1–6] complexed dataset, special care was given in refinement to avoid over parametrisation. MR placed domain positions were first refined using jelly body restraints and conservative weighting (0.0002) between X-ray and geometric restraints. B-factors were also further set to a constant value and only TLS group B factor refinement was performed on the structure. To optimize glycan geometry the final refinement step was performed in PHENIX where the same TLS groups were used and a single grouped B-factor per chain was refined with conservative X-ray/geometry weighting (wxc of 0.05) and optimization of X-ray/ADP weighting. Also here starting model, NCS and secondary structure restraints were used. *MOLPROBITY*[96] was used for structure validation. The composite OMIT map was calculated in *PHENIX* using the "simple" method[97].

### Structural analyses

Structural analyses were performed using various relevant programs. Interface properties and buried surface areas were determined for analysis with the *jsPISA*[47] server. Hydrophobic surface representation coloring was obtained using the YRB coloring scheme[98]. Electrostatic surface properties at pH 7.4 were obtained using the *PDB2PQR*[99] and *APBS*[100] webservers. Conservation analyses were performed using *CONSURF*[101] with curated sequence lists retrieved for vertebrate orthologues from UNIPROT[102] database and chosen paralogues from

the NCBI database[103]. For paralogue conservation analyses, sequences of L1 and contactin paralogues from human, mouse, chicken, xenopus and zebrafish were selected. Figures were generated with *PyMol* (Schrödinger), and the *ILLUSTRATE*[104] webserver. Sequence alignment of L1 mouse paralogues highlighting architecture was prepared using the *ESPript* webserver[105].

### Size exclusion chromatography and multi-angle light scattering

SEC analysis was performed on neurofascin 155[Ig1–6] HM and CG wt, and Thr216Ala HM samples to characterize monomer dimer exchange through peak shift. Purified samples (1–100 μM) were injected onto a Superdex200 10/300 increase (GE Healthcare) column equilibrated in SEC buffer and separated with a flow rate of 0.75 ml min$^{-1}$. For molecular weight characterization of neurofascin 155[Ig1–6] HM and Thr216Ala HM samples, light scattering measurements were performed using a miniDAWN TREOS multi-angle light scattering detector (Wyatt), connected to a differential refractive index monitor (Shimadzu, RID-10A) used for protein concentration quantification. Collected chromatograms were analyzed and processed using ASTRA6 software (Wyatt, using a calculated dn/dc value of 0.182 ml g$^{-1}$, determined from dn/dc of 0.188 and 0.145 for the protein and glycan parts respectively, and 8% glycosylation estimated from crystallographically confirmed glycosylation sites). Instrument calibration was assessed by injection of 5 mg ml$^{-1}$ monomeric conalbumin (Sigma-Aldrich), using in this case a dn/dc value of 0.185 ml g$^{-1}$.

### Surface plasmon resonance imaging

Contactin 1 SPR ligand constructs subcloned in pUPE107.62 (cystatin secretion signal peptide, C-terminal biotin acceptor peptide-His6 tag) were biotinylated in HEK293 cells by co-transfection with E. coli BirA biotin ligase with a sub-optimal secretion signal (in a pUPE5.02 vector), using a DNA ratio of 9:1 (sample:BirA, m/m). Sterile biotin (100 μl of 1 mg/ml HEPES-buffered biotin per 4 ml HEK293 culture) was supplemented to the medium. Contactin 1 ligand samples were purified by Ni Sepharose excel (GE Healthcare) affinity chromatography, with purity evaluated by SDS-PAGE and Coomassie staining. Continuous flow microspotting was used to deposit an array of c-terminally biotinylated proteins on a P-STREP SensEye® (Ssens) chip using a Continuous Flow Microspotter (CFM, Wasatch Microfluidics) with an 8 × 6 format. SEC buffer with 0.005% Tween-20 was used as a spotting buffer and the spotted chip surface was quenched using 1 mM biotin in SEC buffer. C-terminal coupling of ligands to the chip ensured a native-like topology. Surface plasmon resonance imaging experiments were performed on a MX96 SPRi instrument (IBIS Technologies). Analytes in SEC buffer were flown over the sensor chip, with SEC buffer with 0.005% Tween-20 used as running buffer, and 2 M MgCl 25 mM MES pH 5 used as regeneration buffer. During measurement, temperature was kept constant at 25 °C. The data were analyzed using *SPRINTX* (IBIS Technologies) and *PRISM* (Graphpad). The signal was corrected by subtraction of a reference signal using reference regions with no ligand deposited. Response units based on averaged response signal at equilibrium, i.e., between 300 and 380 s of association phase were plotted against the analyte concentration and modeled with a 1:1 Langmuir binding model to calculate the $K_D$ and the maximum analyte binding ($B_{max}$). The theoretical $B_{max}$ was determined from the amount of ligand deposited on the sensor surface and corrected for the difference in molecular weight between the ligand and the analyte. Interaction experiments shown in Supplementary Fig. 1e–g were not quantified.

### Small angle X-ray scattering

Batch SAXS experiments were carried out at the DLS beamline B21 operating at an energy of 12.4 keV and using a sample-to-detector (Eiger 4 M, Dectris) distance of 4.01 m. Scattering of pure water was used to calibrate the intensity to absolute units. Data reduction was performed automatically using the *DAWN*[106] pipeline. Frames were averaged after manual inspection for radiation damage, scattering of SEC buffer was subtracted, and intensities were normalized by concentration. Data were analyzed in *PRIMUS*[107] and results were plotted in *EXCEL* (Microsoft). Expected monomer molecular weights for oligomerization analyses were estimated from sequence derived molecular weight, adding 1.5 kDa or 2 kDa per crystallographically confirmed glycosylation sites for high mannose and complex glycan material, respectively. $P(r)$ analyses were performed on low concentration samples to estimate $d_{max}$ of monomer. Values for higher concentration samples were not determined (n.d) given the obvious non monodispersity of samples. SAXS models for contactin 1[Ig1–6] monomer, neurofascin 155[Ig1–6] monomer and dimer, (all high-mannose versions) were prepared by modeling high mannose glycans trees (two GlcNAc and five Mannose residues) at glycosylation sites observed in the crystal structures using geometric restraints in *COOT*[94]. Contactin 1[fe] model was prepared by aligning and joining contactin 1[Ig1-6] with mouse homology models for contactin 1[FnIII-3], contactin 1[Ig5-FnIII2] and contactin 1[FnIII4] generated by *PHYRE2*[49] with >90% confidence. Relative orientation of domains was not altered to improve the fit of the SAXS model, albeit the orientation of contactin 1 FnIII4 was placed arbitrarily and avoiding clashes given the lack of prior information for the FnIII3-4 connection. For neurofascin 155[Ig1–6] chain A of the crystal structure was used. Predicted scattering, fit to experimental scattering data, and Rg of the models were calculated using the *FoXS*[108] webserver.

### Analytical ultracentrifugation

SV-AUC experiments were carried out in a Beckman Coulter Proteomelab XL-I analytical ultracentrifuge with An-60 Ti rotor (Beckman) at 42,000 or 50,000 revolutions per minute (r.p.m.). Contactin 1[Ig1–6] high mannose at 9/36 μM, and contactin 1[fe] high mannose/complex glycans at 5/22 μM were measured in SEC buffer at 20 °C. Contactin 1[fe] Leu279Arg high mannose at 5/22 μM were measured in SEC buffer supplemented with cOmplete™, Mini, EDTA-free protease inhibitor cocktail at 20 °C. Either 12 mm (5/9 μM sample) or 3 mm (22/36 μM sample) centerpieces with sapphire windows were used. Absorbance was determined at 280 nm using buffer as a reference. A total of 500 scans per cell were collected and analyzed with *SEDFIT*[109] version 16.1c (oct 2018). A continuous c(s) distribution model was fitted to the data, with a resolution of 200 in a sedimentation coefficient range from 0–20 S. The frictional ratio, the baseline and for most of the fits the meniscus were floated in the fitting and the bottom remained fixed. Owing to absorbing remnants arising from protease inhibitors in the mutant contactin 1 sample causing a peak close to $S = 0$, to compare occurrence of higher order species across samples, abundances based on the peak area of oligomer species were determined relative to the normalized area of the monomer peak.

### Native mass spectrometry

Native MS experiments were performed on a modified LCT time-of-flight instrument (Waters). Protein samples were buffer exchanged to 150 mM ammonium acetate (pH 7.5) in six consecutive dilution and concentration steps at 4 °C using Amicon Ultra centrifugal filters with a 10 kDa molecular weight cutoff (Merck). Concentrations of the main stock solutions were determined by measuring the absorbance at 280 nm using a NanoDrop 1000 spectrophotometer (NanoDrop Technologies). Samples were diluted to a concentration of 2.5 μM before analysis, followed by loading into gold-coated borosilicate capillaries (prepared in house) for direct infusion from a static nanoelectrospray ionization source. Data were processed in MassLynx V4.1 (Waters).

### Cell clustering assay, imaging and analysis

K562 cells (kind gift from Dr. Bas van Steensel, Netherlands Cancer Institute) were cultured in RPMI-1640 medium (Gibco), supplemented

with 10% FBS (Gibco) and 1% Penicillin/Streptomycin (Gibco), and grown in a shaking incubator at 37 °C and 5% $CO_2$. Prior to electroporation, K562 cells were collected from culture flasks and centrifuged for 5 min at 300 × g. Cells were then washed in 1x PBS (Gibco), centrifuged for 5 min at 300 × g and resuspended in buffer R (Gibco). Per condition, $2 × 10^6$ cells were incubated with a total amount of 15 μg of DNA for 15 min at room temperature. The ratio of contactin 1$^{WT}$ or neurofascin 155$^{WT}$ plasmid to empty vector was 10:1, while the ratio of contactin 1$^{Mut}$ or neurofascin 155$^{Mut}$ plasmid to empty vector was 1:1. After the incubation, K562 cells, in presence of DNA mixtures, were electroporated with the Neon Transfection System (Thermo Fisher Scientific), using 100 μl tips and the following settings: 1450 V, 10 ms pulse length, and 3 pulses[110]. After electroporation, cells were directly plated in 6-well plates onto 5 ml of pre-warmed RPMI-1640 medium with 10% FBS and either 0.1% DMSO (Sigma-Aldrich) or 10 μM Kifunensine in 0.1% DMSO (Kif; Sigma-Aldrich). Cells were allowed to recover for ~20 h in a shaking incubator at 37 °C and 5% $CO_2$. After recovery, cells were collected and centrifuged for 3 min at 200 g. Cells were then resuspended in assay medium (RPMI-1640 supplemented with 10% FBS) and treated with DNase I (Invitrogen) for 10 min at 37 °C. Cells were once again centrifuged, resuspended in assay medium and passed through a 40 μm cell strainer. Cells were counted using the Countess 3 FL (Thermo Fisher Scientific), and a total of $2 × 10^5$ cells per clustering condition were plated in a 12-well plate in 1 ml of assay medium. Cells were left to cluster for 24 h on a shaking incubator at 37 °C and 5% $CO_2$ and clusters were imaged on an EVOS M5000 microscope with a ×10 objective (0.25 NA; EVOS, Thermo Fisher Scientific), using the EVOS LED GFP and RFP cubes (Thermo Fisher Scientific). For the analysis, GFP and mCherry channels were combined and Otsu thresholding was applied. Regions of interest (ROIs) larger than 50 pixels were identified using Analyze Particles[111]. Rolling ball background subtraction with 50-pixel radius was performed on the individual GFP and mCherry channels, before measuring the area and mean intensities of GFP and mCherry channels of each ROI. A cell cluster was defined as an object three times larger than the determined mean large single cell size calculated from the largest 2.5% in a single image consisting of 1637 single cell ROIs after removal of the largest 10 ROIs. The clustering index was determined as the summed cluster area divided by summed area of all ROIs (clusters + non-clusters) times 100%. The cluster size, co-cluster ratio, and clustering index were averaged per image, and data from three independent experiments (5 images per experiment, 15 images total per condition) were analyzed using *Python 3*[112] and *Seaborn*[113] statistical data visualization.

### Statistical analysis

For the comparison of multiple groups, we used a one-way ANOVA followed by a Tukey's multiple comparison test using GraphPad Prism (GraphPad Software, San Diego, California USA). Differences between conditions were considered significant when $p < 0.05$ (*$p < 0.05$, **$p < 0.01$, ***$p < 0.001$) (Supplementary Table 5). In all figure legends, N indicates the number of independent experiments, and n indicates the number images analyzed. Data are represented as mean values ± SEM.

### Reporting summary

Further information on research design is available in the Nature Research Reporting Summary linked to this article.

## Data availability

Coordinates and structure factors for contactin 1$^{Ig1-6}$–neurofascin 155$^{Ig1-6}$, neurofascin 155$^{Ig1-6}$, contactin 1$^{Ig1-6}$ have been deposited in the Protein Data Bank with accession numbers 7OL4 (contactin 1$^{Ig1-6}$–neurofascin 155$^{Ig1-6}$ complex), 7OK5 (neurofascin 155$^{Ig1-6}$), and 7OL2 (contactin 1$^{Ig1-6}$). All SAXS data have been deposited at the small angle scattering databank (SASBDB) with the accession codes: SASDL66 (neurofascin 155$^{Ig1-6}$ HM 37.2 μM), SASDL76 (neurofascin 155$^{Ig1-6}$ HM 73.5 μM), SASDL86 (neurofascin 155$^{Ig1-6}$ HM 19.7 μM), SASDL96 (neurofascin 155$^{Ig1-6}$ HM 9.5 μM), SASDLA6 (neurofascin 155$^{Ig1-6}$ HM 5.1 μM), SASDLB6 (neurofascin 155$^{Ig1-6}$ HM 2.7 μM), SASDLC6 (neurofascin 155$^{Ig1-6}$ HM 1.3 μM), SASDLD6 (neurofascin 155$^{Ig1-6}$ Thr216Ala HM 109.3 μM), SASDLE6 (neurofascin 155$^{Ig1-6}$ Thr216Ala HM 21.9 μM), SASDLF6 (neurofascin 155$^{Ig1-6}$ Thr216Ala HM 5.5 μM), SASDLG6 (neurofascin 155$^{Ig1-6}$ Thr216Ala HM 1.1 μM), SASDLH6 (contactin 1$^{Ig1-6}$ CG 24.6 μM), SASDLJ6 (contactin 1$^{Ig1-6}$ CG 12.3 μM), SASDLK6 (contactin 1$^{Ig1-6}$ CG 6.2 μM), SASDLL6 (contactin 1$^{Ig1-6}$ CG 3.1 μM), SASDLM6 (contactin 1$^{Ig1-6}$ HM 21.8 μM), SASDLN6 (contactin 1$^{Ig1-6}$ HM 10.9 μM), SASDLP6 (contactin 1$^{Ig1-6}$ HM 5.5 μM), SASDLQ6 (contactin 1$^{Ig1-6}$ HM 2.7 μM), SASDLR6 (contactin 1$^{fe}$ CG 20 μM), SASDLS6 (contactin 1$^{fe}$ CG 10 μM), SASDLT6 (contactin 1$^{fe}$ CG 5 μM), SASDLU6 (contactin 1$^{fe}$ CG 2.5 μM), SASDLV6 (contactin 1$^{fe}$ HM 20.5 μM), SASDLW6 (contactin 1$^{fe}$ HM 11.1 μM), SASDLX6 (contactin 1$^{fe}$ HM 5.5 μM), SASDLY6 (contactin 1$^{fe}$ HM 2.4 μM). Source data are provided with this paper.

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

## Acknowledgements

We thank the staff of the DLS beamlines I03 and I24 for help with X-ray diffraction data collection and of beamline B21 for help with SAXS data collection. L.M.P.C. thanks Nick Pearce, Jitse van der Horn, and Gijs van der Schot, for the instructional conversations regarding crystallography. K562 cells were a kind gift from Dr. Bas van Steensel at Netherlands Cancer Institute. This project has received funding from the European Research Council (ERC) under the European Union's Horizon 2020 research and innovation program with grant agreement No. 677500 (to B.J.C.J.). D.H.M. acknowledges support from Parents in KIND grant, sponsored by the Kavli Institute of Nanoscience, the Department of Bionanoscience in Delft, and the NWO Spinoza Prize. M.A.d.B. and A.J.R.H. acknowledge support from the Netherlands Organization for Scientific Research (NWO) funding the Netherlands Proteomics Centre through the X-omics Road Map program (project 184.034.019).

## Author contributions

B.J.C.J. conceived the project. L.M.P.C. designed experiments with input from B.J.C.J. B.J.C.J., L.M.P.C. and J.C.M.G. cloned various constructs. L.M.P.C. purified recombinant proteins and performed structural and biophysical experiments (X-ray diffraction, SAXS, SEC and SPR) with input from B.J.C.J. L.M.P.C. and B.J.C.J. processed X-ray diffraction data. D.M.E.T.-W. performed SV-AUC experiments and data analysis. M.A.d.B. performed native mass spectrometry experiments and data analysis, with support from A.J.R.H. C.G. and C.P.F. performed cellular experiments and analysis, with support from D.H.M. B.J.C.J. supervised the project. L.M.P.C. and B.J.C.J. analyzed the structural information and wrote the manuscript. All authors commented on the manuscript.

## Competing interests

The authors declare no competing interests.
