## [Peer Review File · Nature Communications]

REVIEWER COMMENTS

Reviewer #1 (Remarks to the Author):

NCOMMS-21-22916A-Z

Review comment:

Chataigner et al., "Structural insights into the contactin 1 – neurofascin 155 adhesion complex"

Contactin1 and neurofascin155 play essential roles in the formation and maintenance of paranodal myelin-axon junctions and their defects cause neuropathies and neurodegenerative disorders. The axonal contactin and glial neurofascin form a complex in trans between the axon and the paranodal loop in the nervous system. In this study, Chataigner et al. structurally characterized the heterophilic interaction between contactin1 and neurofascin155 by crystallography, SPR spectroscopy, analytical ultracentrifugation, native MS, and SAXS and provide novel mechanistic insights into this neurologically important interaction. The results are explicitly presented, although some data need careful interpretation. The following points need to be addressed before publication.

Major points:

1. Overall, the physiological relevance of the detailed structural information is rather unclear owing to the lack of cell-based experiments. The impact of mutations at the intermolecular interface should be assessed in cellular (or near-physiological) contexts, if possible. Experiments using non-neuronal cells (e.g., cell aggregation assay) may be much easier than those using neuronal cells and might reflect the property of the cell-cell adhesion.

2. In Supplementary Figure 1, SPR sensorgrams at higher concentrations show that the SPR signal does not reach a plateau, suggesting that nonspecific binding may occur. Both the theoretical B_{max} (maximum analyte binding) and calculated B_{max} should be presented to guarantee the reliability of the SPR analysis.

3. The crystal structure of the heterophilic complex suggests sugar chain-mediated interactions, although the electron density of the sugar chains is not well resolved due to the moderate resolution. The contribution of the sugar chain-mediated interactions to the total binding should be estimated by analyzing mutants deficient in glycosylation (or samples treated with glycosidases).

Minor points:

1. The terms "FnIII" and "FNIII" are used. Please use either one.

2. The term "fe" may mean "full ectodomain" but is not defined.

3. The shapes of neurofascin155 and contactin1 are expressed as "hoe" and "sickle", respectively, while the position of the "blade" is reversed between them and somehow confusing. Such an analogy may be unnecessary.

4. In Figure 2b and Supplementary Figure 3a, Pro28 and Val439 of neurofascin155 are not labeled.

5. I could not understand what an "untampered" shape of Kratky plot means. Is the word "untampered" commonly used to describe the shape of the plot?

Reviewer #2 (Remarks to the Author):

The manuscript entitled "Structural insights into the contactin 1 – neurofascin 155 adhesion complex" by L.M.P. Chataigner et al. provides a valuable contribution to the understanding of the molecular basis of the contactin 1 – neurofascin 155 complex formation. The presented functional analysis is based mainly on three crystal structures, the very low resolution complex between contactin 1 and neurofascin 155 as well as structures of its individual (unliganded) components resolved to low and moderate (low) resolution. The authors could show that both the heterodimer formed by mouse contactin 1 Ig1-6 and neurofascin 155 Ig1-6 as well as the neurofascin 155 Ig1-6 homodimer are formed via Ig1-2 interactions. Hence both homo and hetero dimers share the similar binding mode as described for the human contactin 2 Ig1-4 and neurofascin 186 Ig1-4 horseshoe homodimers. Furthermore, the authors conclude that all neurofascin isoforms could likely bind contactin 1 using the mentioned Ig1-Ig2 interface and that the whole process is regulated by glycosylation.

It is a very interesting and nicely illustrated manuscript. It provides not only a significant contribution to the understanding of heterodimeric complex formation between contactin 1 and neurofascin 155 but also provides a platform for the development of potential therapeutic agents targeting autoimmune neuropathies and most likely other neurodegenerative diseases.

I am convinced that the manuscript merits publication, however some major and a few minor concerns should be first clarified.

The major highlight of this manuscript is the contactin 1 – neurofascin 155 complex structure, resolved to 4.8 Å resolution (anisotropic). The low resolution itself challenges the macromolecular refinement process and in combination with anisotropic data it becomes very challenging. I have some major concerns related to the refinement process as the current description and missing details implicate a severe model bias what might undermine the reliability of the structure. I would like to state that even if the refinement and/or model rebuilding has not been performed in an optimal way, it can and SHALL be improved. I hope that my comments would be of help.

1. How was the quality of PHYRE2 generated models validated? Did authors use omit maps (any kind) to validate 3D models of Ig-5 and Ig-6 domains obtained by homology modeling? If not – how were the models validated? What was the sequence identity and similarity between the templates and target sequences? This is a crucial question as off register errors occur in homology models (depending on sequence identity of course) and are not easy to spot at medium low and low resolution (3 or 3.9 Å). Refinement strategy of the Cntn1-Ig6 structure will not be commented, please see the paragraph describing concerns for the complex structure.

2. Data processing has been performed using a xia2 pipeline. This could be acceptable for neurofascin 155 and contactin 1 datasets, although the later one exhibits elevated Rmerge (Table 1) most likely due to helical oscillation mode of data acquisition. However a special care should have been taken concerning the complex data set. Are the two measured data sets originating from the same crystal or two crystals have been used? Were the unit cell constants similar between these two data sets (isomorphism)? According to me, a better strategy to perform merging of data sets would be to use XSCALE program and check the merging statistics. I am sure xia2 has such a possibility as well. One could also try to reprocess the "second" data set using the known unit cell constants obtained from the "first/better" data set and allow for "refinement/changes" of the unit cell parameters only during the CORRECT step in XDS (this trick could make the unit cell constants more similar,

assuming of course that they do not differ too much). A careful processing and merging of diffraction images could have a very strong and positive influence on the quality of structure, electron density maps and structure interpretation. An additional suggestion – you could try to use Autoproc as it contains STARANISO module. I would strongly recommend to reprocess the data and provide merging statistics as reported by XSCALE.

3. Looking at the crystallographic Table 1 one can see that for the complex structure number of reflections is lower than number of atoms. By refinement of coordinates only (3 times number of atoms) plus 24 TLS groups (24 x 21) the parameters to observations ratio is really low. Hence, refinement using Refmac5, even with jelly body restraints (I assume only initially), is far away from being optimal. One could consider refinement in torsional space in combination with DEN (Deformable Elastic Network, implemented in PHENIX and CNS) with a fixed B value corresponding to Wilson B and using one TLS group for each chain. There are additional severe issues with refinement using Refmac5.

- Refmac5 does not use measured intensities but structure factor amplitudes "F". This causes a problem if the F's have been generated using CCP4 suite as the applied French-Wilson treatment of the intensities will result in systematically over-estimated amplitudes in the weak directions of diffraction. STARANISO does that conversion properly. Unfortunately description of data treatment is rather scarce hence it is not possible to state it. More details should be provided.

- An additional issue is the used electron density map for manual model adjustments using Coot. Refmac5, by default, provides the so-called "fill-in" maps with not observed amplitudes "replaced" with the structure-factor amplitudes $D|F_{calc}|$ calculated from the model. This is likely to result in highly undesirable strong model bias if not handled correctly, in particular when isotropic completeness is so low as in case of the complex structure. I would strongly recommend to perform re-refinement of the complex structure against re-processed diffraction data using PHENIX or Buster TNT. Instead of using jelly-body restraints one could use DEN refinement which is intended to be used for low resolution crystal structures. The authors should also provide Figures with unbiased electron density maps showing the important structural features discussed in the paper (not a standard 2mFo-DFc map). PHENIX provides not filled electron density maps and allows to generate different kinds of omit maps (less biased). It is understandable that the authors will present a B-factor sharpened map. My question is – how was the sharpening assessed? PHENIX has an automated program to assess the best sharpening parameters.

4. The Nfasc-Ig6 and Cntn1-Ig6 seem to have a very regularized geometry (RMSD for bonds and angles are very low). Were the automated weight adjustment options used during refinement in PHENIX? How did you cope with over-fitting in case of Cntn1-Ig6 structure (3.9 Å resolution, ratio of observations to parameters is ~ 0.5)?

5. The authors performed SAXS experiments and compared the homology modeled contactines with the experimental curves. The homology modeling has been performed using a "black box" server Phyre2 based on two structures: contactin 1 Fn1-3 (PDB:5E53) and contactin 3 Ig5-FnIII2 (PDB:5I99). The authors do not report neither the percentage identity between the modeled sequences and the templates nor any "predicted" confidence of obtained homology models. Hence it is difficult to assess the usability of these models. How were the mannose trees modeled? Were these just "copied" from structures presented in this paper? Was the relative orientation of Ig domains altered to improve the fit to SAXS curve? If so – how was it performed? In addition it would be interesting to see the SAXS based ab initio model (if these could be reliably calculated) in order compare it with the ectodomain SAXS model presented in Figure 3.

6. Usually one sticks to one refinement program when doing crystallographic refinement.

Although it is allowed to refine each structure with a different program, often a mixture of programs implicates potential "problems". What was the reason to switch to Refmac5 for refinement of complex structure? With problematic data Refmac5 allows to ignore standard deviations of measured amplitudes and has a very good treatment of twinning. Was that the reason?

7. The model of complex structure (contactin 1 – neurofascin 155) possesses several geometrical (mostly chirality) and electron-density-fit outliers. There are 62 discrepancies between the modelled and reference sequences. These issues implicate severe problems with refinement and model building and must be corrected.

Minor remarks:

Crystallographic Table 1.

The beta angle (in P21 space group) should be reported with at least one decimal place. In general, at this resolution, the shells could be also reported with a bit lower precision (1 decimal place) and should be consistent with the precision describing the assessment of the unit cell constants.

For anisotropic data CC(1/2) is not suitable as a metric. The authors should provide more detailed statistics reported by STARANISO program like: the 'lowest limit', the 'worst limit' as well as the 'best' one. It would be helpful as well (in particular for electron density map interpretation) to report eigenvalues of overall anisotropy tensor and corresponding eigenvectors.

Line 179

"in close apposition" - for a not native speaker I find this term a bit awkward

Line 247

"The Ig2-Ig3 combination" could be most likely more appropriately expressed as an ensemble or even molecular arrangement. Please consider rephrasing throughout the whole manuscript.

Line 251

apposition → approximation, vicinity.

Lines 200:202

It is somehow difficult to recognize the V-turn, the horseshoe and the Ig5-Ig6 handle based on Figures 1 and 2. A simple video with molecules colored as on Figures 1 and 2, provided as a supplementary data, would make it much easier for the reader.

Suppl. Fig2

2mFobs -Dfcalc map has been shown for the Contactin 1 Ig1-6 – neurofascin 155 Ig1-6 interface. How was this map calculated? Is this a "filled" electron density map or a "not filled" one?

The authors should present an omit map or more preferably a kicked map which should be even less biased.

Reviewer #3 (Remarks to the Author):

The manuscript by Chataigner et al presents a structural characterization of the extracellular moieties of two immunoglobulin-like cell adhesion proteins (contactin 1 and neurofascin 155) and the structural basis of their interaction. These proteins play important roles in the nervous system. The structures of individual fragments of contactin 1 and neurofascin 155 were solved to 3.9 and 3.0 Å respectively, which allows for a reasonable modeling of structural details. These proteins were further characterized by SAXS. The crystal structure of the contactin 1/neurofascin 155 complex is also presented, that structure was solve to low resolution (4.8 Å in the best diffracting direction). Despite this limitation, it is reasonable that the data allowed the identification of the global arrangement of the two proteins and the identification of contact regions. The information extracted from the structure is correct and in consonance with the resolution, so are the interpretations. The contactin 1 binding interface in neurofascin 155 overlaps with the homo-dimerization interface of neurofascin, providing the basis for the competition between these two interactions. This is the first structural description of the interaction between these two types of proteins and has potential important implications to understand the formation of adhesion contacts in the nervous system. The binding interfaces are conserved in paralogues, suggesting that the contactin 1/neurofascin 155 binding mechanism has a wide implication and may serve to understand the interaction between other contactins and members of the L1 family. Therefore, this work has the potential to make a notable contribution to this field.

Major points

- 1) The interaction mechanism revealed by the crystal structure of the contactin/neurofascin complex is very attractive. It would have been very important to have validated the crystallographic information with functional data. That is, to analyze and eventually confirm in a cellular context that the observed binding interfaces and contacts are important for the interaction. Probing the interaction by mutagenesis combined with assays in cells, which are not unusual in the field, would have provider a very strong support. Even if the resolution of the structure does not allow identifying specific contacts, it is reasonable that it would reveal residues whose mutation might affect the interaction. Related to this, are there any disease-linked mutations in these proteins that target the interaction interface that could give functional support to the structural information?**
- 2) The description of some structural properties of contactin 1 and neurofascin 155, such as the conformational plasticity, are interesting and are analyzed rigorously. Yet, their functional relevance is not addressed. Functional analysis would have been very convenient to assess their contribution to the cell adhesion role of these molecules.**
- 3) The suggested oligomerization of contactin1 based on the presence of linear oligomers in the crystal packing is attractive and may be worth analyzing in detail. Yet, the data of the L279R mutant does not really address its relevance. Analysis of the L279R mutant suggests that the crystal interface alpha could be favored to a certain extent in solution. Although, it is intriguing that contactin1-L279R only shows a marginal increase in the abundance of fast sedimenting species compared to the wild type protein. The L279R data do not prove that wild type contactin1 oligomerizes forming similar arrays as those observed in the crystal. It cannot be excluded that L279R could be stabilizing an association that is not relevant in**

solution nor in vivo. The key questions would be whether the contactin 1 interactions and/or oligomerization observed in the crystal occur also on the cell membrane and if they have any role in the adhesive function of contactin1. Unfortunately, the work does not address these issues.

4) Regarding the SPR analysis of the interactions, I believe that the data points in the binding curves shown in Fig 1b are the (normalized) maximal signal during the binding phase. If so, they should be the signals once that the binding curve reaches a plateau (steady state). The SPR plots in supplementary Fig 1 do not reach a plateau. Please, describe in detail the method of analysis of SPR data. If non-steady state values have been used, please, make that clear in the text and include an explanation why the authors consider that the analysis is still valid (or highlight the limitations).

Minor points

5) The analysis of the dimerization of neurofascin155 Ig1-6 by SEC should be taken with caution. The displacement of the elution peak cannot be assigned specifically to dimerization unless the MW would be measured, for example by (SEC-)MALS. On the other hand, analysis of the homo-association by SAXS should yield information about the mean average weight of the species. Yet, instead of R_g (which depends on size and shape) (Fig 2e), plotting the $I(0)$ versus concentration would be more adequate, because $I(0)$ forward scattering is directly related to the effective molecular mass. Actually, $I(0)$ values and the derived molecular masses have been estimated (Suppl Table 1), why not plotting them?

6) Related to the R_g values; how do the R_g values estimated from the 3D structures of neurofascin155 monomer and dimer compare to the experimentally derived R_g values?

7) In the dimensionless Kratky plot in Fig 2e, the data of the T216A mutant above ~ 10 qRg seems to be basically noise. Please, consider cutting the plot to a reasonable limit.

8) In the SPR graphs in supplementary Fig 1, please mark the beginning of the binding step, the dissociation step, and (what it seems to be) the regeneration steps. Otherwise the non-expert reader would have trouble understanding these data.

9) In the sedimentation velocity analysis, $c(s)$ vs s plots in Fig S5, what is the rationale to assign the peaks with higher s -values to dimers and trimers? The aggregation state cannot be derived only from the s -value. If that is how it the association states have been assigned, the peaks in the $c(s)$ vs s plot should not be referred as "dimer" and "trimer". Also, please improve the quality of this figure. The axis labels are too small to be read.

Figures are nicely prepared. The following comments and suggestions are intended to improve their clarity.

10) It would be very informative to add a second 90-degrees turned view in some molecular figures, such as Fig 2a, Fig 3a and Fig 3g. The single view shown makes it very difficult to appreciate the 3D shape of the molecules.

11) The labeling of domains in some figures of molecular structures, such as Fig 2a (neurofascin 155 Ig1-6 structure) and Fig 3a (contactin Ig1-6) relies exclusively in the color code. Color blind people would have trouble identifying of each domain, and probably the general reader too. Placing the labels (Ig1, Ig2 ...) next to each domain would be

clearer (such as in Fig S3b). Also the use of similar color in adjacent domains makes difficult to distinguish the individual domains (for example Ig2 and Ig3 have very similar colors, same for Ig5 Ig6). Please, consider using more different colors in adjacent domains.

12) The use of very similar colors (blue and bluish-green) in Fig 2a (right structure), Fig 2c, the chromatograms in Fig 2d, the SAXS data in Fig 2e, and Fig S2 e,g,h, makes it very difficult to distinguish the different parts of the structures or datasets.

13) The labeling in several figures is too small and very difficult to read. This is particularly problematic for superindices, such as "Ig1-6", "fe", exponentials of molar concentrations, etc. Also the use of very light (for example yellow) font colors makes it difficult to read. Please,

14) In Fig 2d, it would help to expand the scale of the elution volume so that the peaks do not appear so narrow (there are no visible peaks before 11 ml or after 15 ml). Also the insert plots are redundant (and do not show the values of the axis).

15) The figure legends seem far too short. It would be very helpful to explain the data or structures in more detail.

REVIEWER COMMENTS TO NCOMMS-21-22916A-Z

We would like to thank all the reviewers for their constructive comments and suggestions. Below we detail our response to each comment.

Reviewer #1:

Review comment:

Chataigner et al., “Structural insights into the contactin 1 – neurofascin 155 adhesion complex”

Contactin1 and neurofascin155 play essential roles in the formation and maintenance of paranodal myelin-axon junctions and their defects cause neuropathies and neurodegenerative disorders. The axonal contactin and glial neurofascin form a complex in trans between the axon and the paranodal loop in the nervous system. In this study, Chataigner et al. structurally characterized the heterophilic interaction between contactin1 and neurofascin155 by crystallography, SPR spectroscopy, analytical ultracentrifugation, native MS, and SAXS and provide novel mechanistic insights into this neurologically important interaction. The results are explicitly presented, although some data need careful interpretation. The following points need to be addressed before publication.

Major points:

1. Overall, the physiological relevance of the detailed structural information is rather unclear owing to the lack of cell-based experiments. The impact of mutations at the intermolecular interface should be assessed in cellular (or near-physiological) contexts, if possible. Experiments using non-neuronal cells (e.g., cell aggregation assay) may be much easier than those using neuronal cells and might reflect the property of the cell-cell adhesion.

We thank the reviewer for the suggestion of adding cell aggregation experiments. We have now performed cell aggregation experiments that probe the properties of contactin 1 – neurofascin 155 adhesion and that validate our structural findings. In these new experiments we show that *i*, contactin 1 expressing cells co-cluster with neurofascin 155 expressing cells *ii*, this cell co-clustering is disrupted by mutations in the contactin 1 – neurofascin 155 interface we observe in the crystal, *iii*, neurofascin 155 expressing cells cluster in a homophilic manner, *iv*, this homophilic clustering is prevented by mutations in the dimerization interface, *v*, contactin 1 expressing cells do not cluster in a homophilic manner, and *vi*, homophilic neurofascin 155 cell clustering does not prevent heterophilic contactin 1 – neurofascin 155 cell clustering (see new Fig. 5, Sup Fig. 6 and a new paragraph added to last part of the results, lines 370-415). These new cell clustering experiments support our structural and biophysical findings on contactin 1 – neurofascin 155 complex formation, on neurofascin 155 dimerization and lack of contactin homomeric interactions in *trans*. We have updated the abstract, introduction, discussion and methods sections to reflect the addition of the cell-clustering data.

2. In Supplementary Figure 1, SPR sensorgrams at higher concentrations show that the SPR signal does not reach a plateau, suggesting that nonspecific binding may occur. Both the theoretical Bmax (maximum analyte binding) and calculated Bmax should be presented to guarantee the reliability of the SPR analysis.

We have now added an additional SPR experiment with higher ligand density to suppl. fig. 1 (panels k and l), in which we report the theoretical Bmax versus the calculated (modelled) Bmax. This experiment shows that the binding between contactin1 and neurofascin 155 is specific. We have not reported on the theoretical B max due to limitations in the experimental set up. In the set up we used, determination of the theoretical B max relies on quantification of the response of local ligand (RLL). At ligand densities below 400 RU this determination is relatively unreliable due to the offline coupling of the ligand in the Continuous Flow Microspotter (CFM, Wasatch). We here preferred to use lower and equal levels of ligand densities, between ~50 - 200 RU, as these are generally less influenced by aggregation, crowding, mass transport and avidity effects (see also Myszk J. Mol. Recognit. 1999;12:390–408) and to allow comparison between the different ligands. To show that the theoretical Bmax matches with the calculated Bmax we have now added an SPR experiment (supl. fig 1 k and l) in which 1350 RU of ligand (contactin 1^{Ig1-6} HM) were coupled to the surface. The K_D is 0.23 μ M for neurofascin 155^{Ig1-6} HM binding in this experiment, a slightly stronger interaction compared to the K_D of 0.6 μ M determined in the experiments at much lower ligand densities (see supl. fig. 1 a and b). The slight increase in affinity may indicate a weak avidity effect made possible by the higher ligand densities. The calculated Bmax is 647 RU and as the ligand and the analyte have similar masses, the theoretical Bmax is expected to be 1350 RU. This indicates that about 50% of the ligand is occupied by analyte at saturation in this experiment. Possibly not all analyte binding sites on the ligand coupled to the surface (i.e. contactin 1^{Ig1-6}) are available for analyte binding. Of note, in the zipper of contactin 1^{Ig1-6} that we report on, only 50% of the neurofascin binding sites are exposed, we have however not further investigated this hypothetical link with these SPR results. The signal reaches plateau at each concentration in the SPR sensorgram added to supl. fig. 1 (panels k and l). The ratio of the theoretical versus calculated Bmax and the observation that equilibrium is reached after each injection in this new experiment shows that the interaction between contactin 1 and neurofascin 155 is specific.

3. The crystal structure of the heterophilic complex suggests sugar chain-mediated interactions, although the electron density of the sugar chains is not well resolved due to the moderate resolution. The contribution of the sugar chain-mediated interactions to the total binding should be estimated by analyzing mutants deficient in glycosylation (or samples treated with glycosidases).

In our SPR setup, contactin 1 affinity to neurofascin 155 does not seem to depend on whether contactin 1 contains high mannose glycans or complex glycans and neurofascin 155 containing high mannose glycans binds slightly better (2.4 to 3.2 fold) to contactin 1 compared to neurofascin 155 containing complex glycans. Due to the relative minor dependence of glycan chemistry to contactin 1 - neurofascin 155 interaction strength in our SPR setup we have not explored the relative contribution of each glycan to total binding strength any further. In contrast to the SPR experiments, the newly added cell-clustering assays show that high mannose glycans, either on

contactin 1 or on neurofascin 155, but not on both, enable heterophilic cell-cell interaction. This suggests that steric properties of the full-length molecules or structural constraints of the cellular context plays an additional role in transcellular interactions mediated by these molecules. Our current structural data does not inform on the molecular mechanisms that underlies this glycan-dependent interaction in the cellular setting nor do we observe this dependence in the SPR experiments. We do show that mutating the protein-protein interaction interface, observed in the crystal structure, is sufficient to disrupt cell-cell interactions (Fig. 5 and Sup Fig. 6), supporting our structural data and highlighting the importance of this protein-protein interface for *trans* interactions.

Minor points:

1. The terms "FnIII" and "FNIII" are used. Please use either one.

We thank the reviewer for spotting this. We have adapted the manuscript and now use the term FnIII exclusively.

2. The term "fe" may mean "full ectodomain" but is not defined.

We have defined the term in text when first used at line 117: “SAXS analysis reveals that the contactin 1 full ectodomain (fe) is elongated and has flexibility”

3. The shapes of neurofascin155 and contactin1 are expressed as "hoe" and "sickle", respectively, while the position of the "blade" is reversed between them and somehow confusing. Such an analogy may be unnecessary.

As requested by reviewer #2 we have added a movie of the contactin-neurofascin complex and neurofascin dimer that should allow the reader to appreciate the overall architecture of the molecules and we believe should enable the reader to grasp the metaphors we have chosen.

4. In Figure 2b and Supplementary Figure 3a, Pro28 and Val439 of neurofascin155 are not labeled.

We have added these labels to Figure 2b and Supplementary Figure 3a.

5. I could not understand what an "untampered" shape of Kratky plot means. Is the word "untampered" commonly used to describe the shape of the plot?

We thank the reviewer for pointing out this typo from our side. The intended word was “untapered” as in not tapered, with tapered defined as “narrowing gradually towards a point”. We have corrected this in the manuscript.

Reviewer #2:

The manuscript entitled “Structural insights into the contactin 1 – neurofascin 155 adhesion

complex” by L.M.P. Chataigner et al provides a valuable contribution to the understanding of molecular basis of the contactin 1 – neurofascin 155 complex formation. The presented functional analysis is based mainly on three crystal structures, the very low resolution complex between contactin 1 and neurofascin 155 as well as structures of its individual (unliganded) components resolved to low and moderate (low) resolution. The authors could show that both the heterodimer formed by mouse contactin 1 Ig1-6 and neurofascin 155 Ig1-6 as well as the neurofascin 155 Ig1-6 homodimer are formed via Ig1-2 interactions. Hence both homo and hetero dimers share the similar binding mode as described for the human contactin 2 Ig1-4 and neurofascin 186 Ig1-4 horseshoe homodimers. Furthermore, the authors conclude that all neurofascin isoforms could likely bind contactin 1 using the mentioned Ig1-Ig2 interface and that the whole process is regulated by glycosylation.

It is a very interesting and nicely illustrated manuscript. It provides not only a significant contribution to the understanding of heterodimeric complex formation between contactin 1 and neurofascin 155 but also provides a platform for the development of potential therapeutic agents targeting autoimmune neuropathies and most likely other neurodegenerative diseases.

I am convinced that the manuscript merits publication, however some major and a few minor concerns should be first clarified.

The major highlight of this manuscript is the contactin 1 – neurofascin 155 complex structure, resolved to 4.8 Å resolution (anisotropic). The low resolution itself challenges the macromolecular refinement process and in combination with anisotropic data it becomes very challenging. I have some major concerns related to the refinement process as the current description and missing details implicate a severe model bias what might undermine the reliability of the structure. I would like to state that even if the refinement and/or model rebuilding has not been performed in an optimal way, it can and SHALL be improved. I hope that my comments would be of help.

1. How was the quality of PHYRE2 generated models validated? Did authors use omit maps (any kind) to validate 3D models of Ig-5 and Ig-6 domains obtained by homology modeling? If not – how were the models validated? What was the sequence identity and similarity between the templates and target sequences? This is a crucial question as off register errors occur in homology models (depending on sequence identity of course) and are not easy to spot at medium low and low resolution (3 or 3.9 Å). Refinement strategy of the Cntn1-Ig6 structure will not be commented, please see the paragraph describing concerns for the complex structure.

We thank the reviewer for pointing this out and apologize for not describing the Phyre2 modeling more clearly. We have now described in the methods section of the manuscript how the quality of the Phyre2 generated models were assessed and improved by model building (also detailed below). We would like to note that the relatively high solvent content of both crystals, 58% for neurofascin 155 (to 3 Å resolution) and 67% for contactin 1 (to 3.9 Å resolution) and the presence of twofold non-crystallographic symmetry for both crystals, aided the model rebuilding process. Below we detail this further.

The quality of the Phyre2 generated models are >90% as reported by Phyre2:

- contactin 1 Ig5, confidence in the model: 102 residues (100%) modelled at >90% confidence
- contactin 1 Ig6 confidence in the model: 94 residues (100%) modelled at >90% confidence
- neurofascin 155 Ig5, confidence in the model: 97 residues (100%) modelled at >90% confidence
- neurofascin 155 Ig6, confidence in the model: 101 residues (100%) modelled at >90% confidence

Inspection and correction of the register and placement of residues in coot was performed iteratively between refinement runs by comparing the model with the electron density map. The electron density for both the neurofascin 155 and the contactin 1 maps was of sufficient quality to allow rebuilding of the models. In particular bulky residues guided the modelling of the correct register. Readjusting the positioning of residues or correcting the register was done on the following parts and improved the Rwork and Rfree values: neurofascin 155 Ig 5 residues 445-449 and 482-492, neurofascin 155 Ig 6 residues 564-584, contactin 1 Ig 5 residues 415-422, and contactin 1 Ig 6 residues 549-566. Remarkably these refined models of the Ig5 and Ig6 domains of neurofascin 155 and contactin 1 are very similar to recently released structure predictions made with alphafold2 in collaboration with the EMBL: <https://alphafold.ebi.ac.uk/>, with rmsd values of 0.521 and 0.624 Å for neurofascin 155 Ig5 and Ig6, respectively, and rmsd values of 0.803 and 0.635 Å for contactin 1 Ig5 and Ig6, respectively. Please note that our structures were not released at the time the alphafold2 models were generated.

2. Data processing has been performed using a xia2 pipeline. This could be acceptable for neurofascin 155 and contactin 1 datasets, although the later one exhibits elevated Rmerge (Table 1) most likely due to helical oscillation mode of data acquisition. However, a special care should have been taken concerning the complex data set. Are the two measured data sets originating from the same crystal or two crystals have been used? Were the unit cell constants similar between these two data sets (isomorphism)? According to me, a better strategy to perform merging of data sets would be to use XSCALE program and check the merging statistics. I am sure xia2 has such a possibility as well. One could also try to reprocess the “second” data set using the known unit cell constants obtained from the “first/better” data set and allow for “refinement/changes” of the unit cell parameters only during the CORRECT step in XDS (this trick could make the unit cell constants more similar, assuming of course that they do not differ too much). A careful processing and merging of diffraction images could have a very strong and positive influence on the quality of structure, electron density maps and structure interpretation. An additional suggestion – you could try to use Autoproc as it contains STARANISO module. I would strongly recommend to reprocess the data and provide merging statistics as reported by XSCALE.

We apologize for not describing the diffraction data processing of the contactin1-neurofascin 155 crystal in more detail and thank the reviewer for pointing this out. Below we explain how we treated our data.

The two datasets of the complex were collected in succession from the same crystal and are isomorphous as assessed by merging statistics. Unmerged, unscaled reflection data without diffraction cut-off for both datasets obtained from an intermediate step in the xia2 pipeline were combined, using pointless, to generate an unmerged, unscaled combined dataset. We then used the recommended procedure from the STARANISO webserver, providing the unmerged, unscaled, combined dataset to scale and merge this data. By merging the two datasets in this manner a dataset with more unique reflections (due to the higher resolution cut-off as automatically determined by STARANISO) is obtained. We have updated the methods section with the above description. The combined dataset has a total of 15712 unique reflections, which represents ~40% more reflections than dataset 1 (9426) and ~20% more reflections than dataset 2 (12654) as determined by STARANISO. See below the comparison of the dataset statistics, as returned by STARANISO for the two individually processed datasets and the combined dataset.

Dataset1

	Overall	InnerShell	OuterShell
Low resolution limit	75.908	75.908	7.135
High resolution limit	6.097	15.843	6.097
Rmerge (all I+ & I-)	0.241	0.133	1.003
Rmerge (within I+/I-)	0.209	0.110	0.843
Rmeas (all I+ & I-)	0.286	0.162	1.170
Rmeas (within I+/I-)	0.292	0.155	1.178
Rpim (all I+ & I-)	0.152	0.091	0.601
Rpim (within I+/I-)	0.203	0.109	0.821
Total number of observations	33346	3135	3530
Total number unique	9426	941	943
Mean(I)/sd(I)	4.0	8.8	1.3
Completeness (spherical)	59.2	100.0	15.9
Completeness (ellipsoidal)	88.3	100.0	53.7
Multiplicity	3.5	3.3	3.7
CC(1/2)	0.933	0.923	0.513
Anomalous completeness (spherical)	57.6	95.3	16.3
Anomalous completeness (ellipsoidal)	85.9	95.3	54.0
Anomalous multiplicity	1.8	1.8	1.9
CC(ano)	-0.204	-0.383	0.025
DANO /sd(DANO)	0.778	0.876	0.778

Dataset2

	Overall	InnerShell	OuterShell
Low resolution limit	75.867	75.867	6.169
High resolution limit	5.426	16.135	5.426
Rmerge (all I+ & I-)	0.188	0.135	0.862
Rmerge (within I+/I-)	0.163	0.109	0.710
Rmeas (all I+ & I-)	0.223	0.165	1.023
Rmeas (within I+/I-)	0.227	0.153	0.997
Rpim (all I+ & I-)	0.119	0.093	0.547
Rpim (within I+/I-)	0.158	0.107	0.699
Total number of observations	44998	3003	3111
Total number unique	12654	902	905
Mean(I)/sd(I)	4.6	11.3	1.4
Completeness (spherical)	55.8	99.9	12.6
Completeness (ellipsoidal)	89.6	99.9	58.3
Multiplicity	3.6	3.3	3.4
CC(1/2)	0.902	0.867	0.561
Anomalous completeness (spherical)	54.2	95.5	12.6
Anomalous completeness (ellipsoidal)	87.0	95.5	56.9
Anomalous multiplicity	1.8	1.8	1.7
CC(ano)	-0.359	-0.437	0.043
DANO /sd(DANO)	0.771	1.002	0.890

Combined

	Overall	InnerShell	OuterShell
Low resolution limit	75.887	75.887	5.352
High resolution limit	4.795	16.857	4.795
Rmerge (all I+ & I-)	0.312	0.080	1.926
Rmerge (within I+/I-)	0.304	0.079	1.803
Rmeas (all I+ & I-)	0.336	0.088	2.068
Rmeas (within I+/I-)	0.356	0.093	2.103
Rpim (all I+ & I-)	0.125	0.034	0.752
Rpim (within I+/I-)	0.183	0.048	1.077
Total number of observations	111848	5128	5908
Total number unique	15712	784	786
Mean(I)/sd(I)	4.5	12.8	1.3
Completeness (spherical)	48.2	99.6	8.7
Completeness (ellipsoidal)	88.5	99.6	73.5
Multiplicity	7.1	6.5	7.5
CC(1/2)	0.993	0.995	0.399
Anomalous completeness (spherical)	47.2	94.9	8.9
Anomalous completeness (ellipsoidal)	86.7	94.9	73.5
Anomalous multiplicity	3.7	3.6	3.8
CC(ano)	0.022	0.275	-0.010
DANO /sd(DANO)	0.755	0.608	0.845

3. Looking at the crystallographic Table 1 one can see that for the complex structure number of reflections is lower than number of atoms. By refinement of coordinates only (3 times number of atoms) plus 24 TLS groups (24 x 21) the parameters to observations ratio is really low. Hence, refinement using Refmac5, even with jelly body restraints (I assume only initially), is far away from being optimal. One could consider refinement in torsional space in combination with DEN (Deformable Elastic Network, implemented in PHENIX and CNS) with a fixed B value corresponding to Wilson B and using one TLS group for each chain. There are additional severe issues with refinement using Refmac5.

- Refmac5 does not use measured intensities but structure factor amplitudes “F”. This causes a problem if the F’s have been generated using CCP4 suite as the applied French-Wilson treatment of the intensities will result in systematically over-estimated amplitudes in the weak directions of diffraction. STARANISO does that conversion properly. Unfortunately description of data treatment is rather scarce hence it is not possible to state it. More details should be provided.

- An additional issue is the used electron density map for manual model adjustments using Coot. Refmac5, by default, provides the so-called “fill-in” maps with not observed amplitudes “replaced” with the structure-factor amplitudes $D|F_{\text{calc}}|$ calculated from the model. This is likely to result in highly undesirable strong model bias if not handled correctly, in particular when isotropic completeness is so low as in case of the complex structure. I would strongly recommend to perform re-refinement of the complex structure against re-processed diffraction data using PHENIX or Buster TNT. Instead of using jelly-body restraints one could use DEN refinement which is intended to be used for low resolution crystal structures. The authors should also provide Figures with unbiased electron density maps showing the important structural features discussed in the paper (not a standard $2mF_o-DF_c$ map). PHENIX provides not filled electron density maps and allows to generate different kinds of omit maps (less biased). It is understandable that the authors will present a B-factor sharpened map. My question is – how was the sharpening assessed? PHENIX has an automated program to assess the best sharpening parameters.

We appreciate the reviewers concerns with the crystallographic refinement and are happy to provide some clarifications about the data treatment prior to and during the refinement procedure. In addition, as suggested by the reviewer, we have now elaborated on the refinement process in the methods section of the manuscript.

- For the refinement of low-resolution structures, the availability of higher resolution structures, partial structures and combined use of tight geometric restraints and jelly body restraints is particularly powerful to stabilize the refinement process. The relatively small Rwork/Rfree gap of 3% for the contactin1-neurofascin155 complex indicates that it is unlikely we have over parameterised our model during the refinement of the structure.
- We have used the recommended procedure for data treatment by STARANISO. The conversion of intensities to structure factor amplitudes was done in STARANISO. As

suggested by the reviewer, we have now elaborated on this procedure in the methods section of the manuscript.

- The contactin1-neurofascin155 dataset used for refinement and map generation did not contain any 'unobserved' and 'unobservable' reflections, we had removed these h k l's from the dataset file prior to refinement and model building as to prevent the filling in of these reflections as the reviewer describes. We now describe this procedure in the method section of the manuscript. In short, SFtools was used to remove unobserved and unobservable reflections, i.e those that lie outside the anisotropic cut-off surface, from the file output by STARANISO. Please note that this is the recommended procedure as suggested in the STARANISO manual. The result is that REFMAC has only filled in the 'observable' missing reflections with DF_{calc} that lie inside the diffraction cut-off surface. There were only 16 of these 'observable' missing reflections, compared to 15712 observed reflections (see table "Combined" above). This has minimized model bias in the maps calculated for model building.
- The optimum map-sharpening factor was obtained by manual assessment. We manually assessed the optimal B-factor sharpening in coot by calculating sharpened maps with Bfactor sharpening ranging from -200 to -50. A sharpening of -150 \AA^2 gives the best compromise, as assessed manually, between sharper higher-resolution features and introducing too much noise. Please note that the automatic calculation of the sharpening factor in coot sets it to the maximum (-200). We preferred to use the sharpening somewhat conservatively to limit the introduction of too much high-resolution noise. As suggested by the reviewer we have now also added an OMIT map of the important contactin 1 – neurofascin 155 interface region (Sup Fig. 2j) that was calculated by the "simple" method in PHENIX.

4. The Nfasc-Ig6 and Cntn1-Ig6 seem to have a very regularized geometry (RMSD for bonds and angels are very low). Were the automated weight adjustment options used during refinement in PHENIX? How did you cope with over-fitting in case of Cntn1-Ig6 structure (3.9 A resolution, ratio of observations to parameters is ~ 0.5)?

Automatic weighting options were used during refinement for both neurofascin 155 Ig1-6 and for contactin 1 Ig1-6. To avoid overfitting, the contactin 1 Ig1-6 structure refinement was performed by applying geometry, secondary structure and NCS restraints. We have updated the methods section in the manuscript to clarify this.

5. The authors performed SAXS experiments and compared the homology modeled contactins with the experimental curves. The homology modeling has been performed using a "black box" server Phyre2 based on two structures: contactin 1 Fn1-3 (PDB:5E53) and contactin 3 Ig5-FnIII2 (PDB:5I99). The authors do not report neither the percentage identity between the modeled sequences and the templates nor any "predicted" confidence of obtained homology models. Hence it is difficult to assess the usability of these models. How were the mannose trees

modeled? Were these just “copied” from structures presented in this paper? Was the relative orientation of Ig domains altered to improve the fit to SAXS curve? If so – how was it performed? In addition it would be interesting to see the SAXS based ab initio model (if these could be reliably calculated) in order compare it with the ectodomain SAXS model presented in Figure 3.

As suggested by the reviewer we have now updated the method section in the manuscript to describe the SAXS analysis and homology modelling in more detail, we apologise for not describing this more clearly initially. The sequence identity between the highest identity template and contactin 1 FnIII1-3 is 77 % and Phyre2 assigns >90% accuracy for 300 (100%) of the residues of the model. The sequence identity between the highest identity template and contactin 1 Ig5-FnIII3 is 46% and Phyre2 assigns >90% accuracy for 397 (99%) of the residues of the model.

Mannose trees were modelled at glycosylation sites supported by electron density, using the carbohydrate/glyco module and geometric restraints in coot. Trees were extended with basic branching with alpha 1-6 and alpha 1-3 mannoses as expected for material produced in a N-acetylglucosaminyltransferase I-deficient (GnTI-) EBNA1-expressing HEK293 cell line.

Relative orientation of domains was not altered to improve the fit of the SAXS model, albeit the orientation of contactin 1 domain FnIII4, the most C-terminal domain, was placed arbitrarily and avoiding clashes given the lack of prior information for the FnIII3-4 connection. We made the deliberate choice to validate the saxs data using our crystal structure and “superposition” homology model over *ab-initio* modelling given the more reliable parametrisation of the crystal structure and “superposition” homology model. We therefor did not attempt *ab-initio* modelling of the SAXS data.

6. Usually one sticks to one refinement program when doing crystallographic refinement. Although it is allowed to refine each structure with a different program, often a mixture of programs implicates potential “problems”. What was the reason to switch to Refmac5 for refinement of complex structure? With problematic data Refmac5 allows to ignore standard deviations of measured amplitudes and has a very good treatment of twinning. Was that the reason?

We used refmac5 for refinement of the complex structure given the powerful implementation of jelly body refinement, which enables refinement to good Rwork/Rfree values of low-resolution data. No treatment of twinning was used as the crystals are not twinned. To correct chirality outliers in the glycans (see next comment) we have now further refined the model of contactin 1 – neurofascin 155 in PHENIX against intensities (as this is the standard option in PHENIX) using the same Rfree set as used in Refmac5. We would like to note the model quality has improved and Rwork and Rfree values are similar comparing Refmac5 and PHENIX refinement.

7. The model of complex structure (contactin 1 – neurofascin 155) possesses several geometrical (mostly chirality) and electron-density-fit outliers. There are 62 discrepancies between the modelled and reference sequences. These issues implicate severe problems with refinement and model building and must be corrected.

We thank the reviewer for pointing out these problems with the contactin1 1- neurofascin 155 deposition in the PDB. To correct the chirality errors the final refinement steps have now been performed in PHENIX using the same Rfree set, and same TLS groups as in Refmac5. Also here, we employed NCS, secondary structure and starting model restraints. We have updated the manuscript and the pdb deposition to reflect these changes. The 62 discrepancies between the model and reference sequence arose due to assigning the incorrect reference sequence from uniprot in the deposition to the PDB. These are therefor not modeling errors, i.e. the sequence of the model was correct. The assignment of the reference sequence has now been corrected in the updated PDB deposition. The reviewer is correct that there are several electron-density-fit outliers. We did minimal adjustments to the model in Coot after rigid body placement to prevent too much deviation from the starting structures that are based on higher resolution data. Most of the outliers are in regions with high B-factor and are likely in the correct position considering the low resolution and anisotropy of the data. To prevent the risk of distorting the model we have not attempted to correct these electron-density-fit outliers.

Minor remarks:

Crystallographic Table 1.

The beta angle (in P21 space group) should be reported with at least one decimal place. In general, at this resolution, the shells could be also reported with a bit lower precision (1 decimal place) and should be consistent with the precision describing the assessment of the unit cell constants.

We now report appropriately for the complex structure the beta angle as 111.8 ° instead of 112 ° for this low-resolution structure. We also report resolution shells with the preferred precision as requested by the reviewer.

For anisotropic data CC(1/2) is not suitable as a metric. The authors should provide more detailed statistics reported by STARANISO program like: the 'lowest limit', the 'worst limit' as well as the 'best' one. It would be helpful as well (in particular for electron density map interpretation) to report eigenvalues of overall anisotropy tensor and corresponding eigenvectors.

As requested by the reviewer, we have added lowest/worst/best limits and eigenvalues and vectors of overall anisotropy tensor to table 1.

Line 179

“in close apposition” - for a not native speaker I find this term a bit awkward

Upon the reviewer's suggestion, we have changed this sentence to:

“The glycan at contactin 1 Asn208 is in near vicinity to Ig2 of neurofascin 155”

Line 247

“The Ig2-Ig3 combination” could be most likely more appropriately expressed as an ensemble or even molecular arrangement. Please consider rephrasing throughout the whole manuscript.

Upon the reviewer’s suggestion, we have re-written where necessary “The Ig2-Ig3 combination” as the “Ig2-Ig3 unit”.

Line 251

apposition → approximation, vicinity.

Upon the reviewer’s suggestion, we have changed this sentence to:

“Glycans at Asn residues 316, 457 and 494 are juxtaposed in the space between the Ig1-Ig4 horseshoe and the Ig5-Ig6 combination”

Lines 200:202

It is somehow difficult to recognize the V-turn, the horseshoe and the Ig5-Ig6 handle based on Figures 1 and 2. A simple video with molecules colored as on Figures 1 and 2 , provided as a supplementary data, would make it much easier for the reader.

As requested by reviewer we have added movies of the contactin 1-neurofascin 155 complex, the neurofascin 155 dimer and the contactin 1 monomer that should allow the reader to appreciate the overall architecture of the molecules.

Suppl. Fig2

2mFobs -DFcalc map has been shown for the Contactin 1 Ig1-6 – neurofascin 155 Ig1-6 interface. How was this map calculated? Is this a “filled” electron density map or a “not filled” one?

The authors should present an omit map or more preferably a kicked map which should be even less biased.

As mentioned above, the 2mFobs-DFcalc map for the Contactin 1 Ig1-6 – neurofascin 155 Ig1-6 complex is not “filled in” as unobserved and unobservable reflections were removed from the dataset to minimize model bias. As suggested by the reviewer we have now also added a figure with an omit map, calculated in Phenix as a composite omit map using the “simple” method (see Afonine, et al., Acta Cryst. (2015) D71, 646-666), to suppl. fig. 2 (panel j). This omit map was not B factor sharpened. The similarity of the 2mFobs-DFcalc map (panel i) and the omit map (panel j) indicates that the model bias is minimal.

Reviewer #3:

The manuscript by Chataigner et al presents a structural characterization of the extracellular moieties of two immunoglobulin-like cell adhesion proteins (contactin 1 and neurofascin 155)

and the structural basis of their interaction. These proteins play important roles in the nervous system. The structures of individual fragments of contactin 1 and neurofascin 155 were solved to 3.9 and 3.0 Å respectively, which allows for a reasonable modeling of structural details. These proteins were further characterized by SAXS. The crystal structure of the contactin 1/neurofascin 155 complex is also presented, that structure was solved to low resolution (4.8 Å in the best diffracting direction). Despite this limitation, it is reasonable that the data allowed the identification of the global arrangement of the two proteins and the identification of contact regions. The information extracted from the structure is correct and in consonance with the resolution, so are the interpretations. The contactin 1 binding interface in neurofascin 155 overlaps with the homo-dimerization interface of neurofascin, providing the basis for the competition between these two interactions. This is the first structural description of the interaction between these two types of proteins and has potential important implications to understand the formation of adhesion contacts in the nervous system. The binding interfaces are conserved in paralogues, suggesting that the contactin 1/neurofascin 155 binding mechanism has a wide implication and may serve to understand the interaction between other contactins and members of the L1 family. Therefore, this work has the potential to make a notable contribution to this field.

Major points

1) The interaction mechanism revealed by the crystal structure of the contactin/neurofascin complex is very attractive. It would have been very important to have validated the crystallographic information with functional data. That is, to analyze and eventually confirm in a cellular context that the observed binding interfaces and contacts are important for the interaction. Probing the interaction by mutagenesis combined with assays in cells, which are not unusual in the field, would have provided a very strong support. Even if the resolution of the structure does not allow identifying specific contacts, it is reasonable that it would reveal residues whose mutation might affect the interaction. Related to this, are there any disease-linked mutations in these proteins that target the interaction interface that could give functional support to the structural information?

We thank the reviewer for the suggestion of adding cellular assays in which we validate the structural findings. This was also requested by reviewer#1. We have now performed cell aggregation experiments that probe the properties of contactin 1 – neurofascin 155 adhesion and that validate our structural findings. This has also been described above, at the first comment from reviewer #1. To reiterate: In the new experiments we show that *i*, contactin 1 expressing cells co-cluster with neurofascin 155 expressing cells *ii*, this cell co-clustering is disrupted by mutations in the contactin 1 – neurofascin 155 interface we observe in the crystal, *iii*, neurofascin 155 expressing cells cluster in a homophilic manner, *iv*, this homophilic clustering is prevented by mutations in the dimerization interface, *v*, contactin 1 expressing cells do not cluster in a homophilic manner, and *vi*, homophilic neurofascin 155 cell clustering does not prevent heterophilic contactin 1 – neurofascin 155 cell clustering (see new Fig. 5, Sup Fig. 6 and a new paragraph added to last part of the results, lines 370-415). These new cell clustering experiments support our structural and biophysical findings on contactin 1 – neurofascin 155 complex

formation, on neurofascin 155 dimerization and lack of contactin homomeric interactions in *trans*. We have updated the abstract, introduction, discussion and methods sections to reflect the addition of the cell-clustering data.

2) The description of some structural properties of contactin 1 and neurofascin 155, such as the conformational plasticity, are interesting and are analyzed rigorously. Yet, their functional relevance is not addressed. Functional analysis would have been very convenient to assess their contribution to the cell adhesion role of these molecules.

We agree with the reviewer that the structural properties of contactin 1 and neurofascin 155, such as the conformational plasticity, may have functional relevance and thank the reviewer for pointing out the rigour of our analysis. It is currently not clear in which functional context the plasticity is important to the molecules. We did not experimentally determine the role of the plasticity in cell adhesion, this would require, possibly challenging, design and biophysical verification of more stable versions of the proteins. While not providing an experimental functional verification of the importance of the plasticity we have however commented, in the discussion, on the possible functional relevance of contactin 1 and neurofascin 155 conformational plasticity in cell adhesion and on bridging different intercellular distances. In addition, we have now added to the discussion that the functional verification of the structural plasticity in the adhesion roles of these molecules awaits further functional verification.

3) The suggested oligomerization of contactin1 based on the presence of linear oligomers in the crystal packing is attractive and may be worth analyzing in detail. Yet, the data of the L279R mutant does not really address its relevance. Analysis of the L279R mutant suggests that the crystal interface alpha could be favored to a certain extent in solution. Although, it is intriguing that contactin1-L279R only shows a marginal increase in the abundance of fast sedimenting species compared to the wild type protein. The L279R data do not prove that wild type contactin1 oligomerizes forming similar arrays as those observed in the crystal. It cannot be excluded that L279R could be stabilizing an association that is not relevant in solution nor in vivo. The key questions would be whether the contactin 1 interactions and/or oligomerization observed in the crystal occur also on the cell membrane and if they have any role in the adhesive function of contactin1. Unfortunately, the work does not address these issues.

The reviewer is correct in that we have not shown experimentally that the contactin 1 zipper that we observe has a physiological role. In the discussion we pointed this out: “It is currently not clear if the zipper we identified plays a role in this setting...”. To further emphasise this, we have now added the following to the discussion: “Finally, secreted contactin 1 plays a role in mediating nodal sodium channel clustering [76] and it is possible that the oligomeric form of contactin 1 has a role in this function, but this has not been experimentally verified.”

4) Regarding the SPR analysis of the interactions, I believe that the data points in the binding curves shown in Fig 1b are the (normalized) maximal signal during the binding phase. If so, they should be the signals once that the binding curve reaches a plateau (steady state). The SPR plots

in supplementary Fig 1 do not reach a plateau. Please, describe in detail the method of analysis of SPR data. If non-steady state values have been used, please, make that clear in the text and include an explanation why the authors consider that the analysis is still valid (or highlight the limitations).

We apologize for not describing the SPR analysis method in more detail. We have now extended the methods section for the SPR analysis with the following: “Response units based on averaged response signal at equilibrium, i.e. between 300 and 380 seconds of association phase were plotted against the analyte concentration and modeled with a 1:1 Langmuir binding model to calculate the K_D and the maximum analyte binding (B_{max}). In experiments where equilibrium is not reached, for example for neurofascin 155^{Ig1-6} HM binding to contactin 1^{fe} CG, the K_D ’s are approximations”. We have added to the legend of supl. fig 1: “In experiments where equilibrium is not reached, for example for neurofascin 155^{Ig1-6} HM binding to contactin 1^{fe} CG (panel d), the K_D ’s are approximations. Nevertheless, the comparison of the four experiments with neurofascin 155^{Ig1-6} HM analyte (panel a) versus the four equivalent experiments with neurofascin 155^{Ig1-6} CG analyte (panel b) consistently shows a weaker affinity of neurofascin 155^{Ig1-6} CG for contactin 1 variants”. In addition, we have added an additional experiment to supl. fig. 1 (panels j and k), that show that equilibrium is reached at higher ligand densities. This experiment is discussed in response to reviewer#1 at comment 2.

Minor points

5) The analysis of the dimerization of neurofascin155 Ig1-6 by SEC should be taken with caution. The displacement of the elution peak cannot be assigned specifically to dimerization unless the MW would be measured, for example by (SEC-)MALS.

On the other hand, analysis of the homo-association by SAXS should yield information about the mean average weight of the species. Yet, instead of Rg (which depends on size and shape) (Fig 2e), plotting the I(0) versus concentration would be more adequate, because I(0) forward scattering is directly related to the effective molecular mass. Actually, I(0) values and the derived molecular masses have been estimated (Suppl Table 1), why not plotting them?

We have now added the requested SEC-MALS analysis in fig 2d and sup table 2. This additional experiment confirms our previous conclusion that the shift in elution volume of neurofascin 155 Ig1-6 is due to dimerization. The MALS determined weight average MW of neurofascin 155 Ig1-6 drops from 102.9 kDa (likely due to a mixture of monomer and dimer present) at 100 μ M injection concentration to 80.5 kDa, a closer match with the mass of a monomer (76 kDa) at 1 μ M injection concentration. Please note that the weight average MW of the mutant neurofascin 155 Ig1-6 barely varies and is 77.4 kDa at 100 μ M injection concentration and 76.2 kDa at 1 μ M injection concentration (see also the added supl table 2).

We appreciate the reviewer suggestion to use the I0 rather than the Rg for our plots given the obvious relationship between I0 and MW. We have now added a plot showing the scaled I0 versus the concentration to supplementary fig 3. We prefer to also show the Rg versus concentration plot

(Fig 2e) as the R_g does not depend on the calibration of the intensity to absolute units and the additional scaling of I_0 for the concentration of the sample to represent the molecular weight.

6) Related to the R_g values; how do the R_g values estimated from the 3D structures of neurofascin155 monomer and dimer compare to the experimentally derived R_g values?

Given the reviewers question we have added estimated R_g values of monomer (R_g of 3.43 nm) and dimer (R_g of 4.59 nm) neurofascin 155 Ig1-6 to supplementary table 1. These values are consistent with the experimental values we report for the WT (3.90-4.98) and mutant (3.58-4.29) neurofascin 155 proteins likely found in monomer-dimer equilibria. Flexibility of the molecules and of the glycan trees in the SAXS experiments may explain the slightly lower estimates (3.43 and 4.59) obtained from the static 3D structures.

7) In the dimensionless Kratky plot in Fig 2e, the data of the T216A mutant above $\sim 10 qR_g$ seems to be basically noise. Please, consider cutting the plot to a reasonable limit.

Upon the reviewer's suggestion we have cut the plot to $10 qR_g$.

8) In the SPR graphs in supplementary Fig 1, please mark the beginning of the binding step, the dissociation step, and (what it seems to be) the regeneration steps. Otherwise the non-expert reader would have trouble understanding these data.

We have now labeled and outlined the association, dissociation and regeneration steps in supplementary figure 1.

9) In the sedimentation velocity analysis, $c(s)$ vs s plots in Fig S5, what is the rationale to assign the peaks with higher s -values to dimers and trimers? The aggregation state cannot be derived only from the s -value. If that is how it the association states have been assigned, the peaks in the $c(s)$ vs s plot should not be referred as "dimer" and "trimer". Also, please improve the quality of this figure. The axis labels are too small to be read.

The peaks in the AUC were assigned as monomer, dimer and trimer based on the S values and the molecular weight calculated using the weight-average best-fit frictional ratio. While these masses may not be accurate for all solution species, together with the much more accurate masses and species determined by native mass spectrometry we are confident of these assignments. We have updated the legend of suppl. Fig. 5 to explain how the assignments were derived and improved the quality of the figures by enlarging the axis labels.

Figures are nicely prepared. The following comments and suggestions are intended to improve their clarity.

10) It would be very informative to add a second 90-degrees turned view in some molecular

figures, such as Fig 2a, Fig 3a and Fig 3g. The single view shown makes it very difficult to appreciate the 3D shape of the molecules.

As requested by reviewer number 2 we have added movies of the contactin-neurofascin complex, the neurofascin 155 dimer and the contactin 1 monomer that should allow the reader to appreciate the overall architecture of the molecules.

11) The labeling of domains in some figures of molecular structures, such as Fig 2a (neurofascin 155 Ig1-6 structure) and Fig 3a (contactin Ig1-6) relies exclusively in the color code. Color blind people would have trouble identifying of each domain, and probably the general reader too. Placing the labels (Ig1, Ig2 ...) next to each domain would be clearer (such as in Fig S3b). Also the use of similar color in adjacent domains makes difficult to distinguish the individual domains (for example Ig2 and Ig3 have very similar colors, same for Ig5 Ig6). Please, consider using more different colors in adjacent domains.

Upon the reviewer's suggestion we have added labels to Fig2a and Fig3a as suggested so that the labelling does not solely rely on color code, this helps to distinguish the individual domains considerably. We thank the reviewer for the suggestion. In addition, the additional movies of all the molecules should allow the reader further means to distinguish the individual domains.

12) The use of very similar colors (blue and bluish-green) in Fig 2a (right structure), Fig 2c, the chromatograms in Fig 2d, the SAXS data in Fig 2e, and Fig S2 e,g,h, makes it very difficult to distinguish the different parts of the structures or datasets.

We have enhanced the difference in colors by changing the tint. As a sidenote, we deliberately chose somewhat similar colors as to enable part of the visual storytelling of the paper highlighting the distinction between the homophilic and heterophilic complexes.

13) The labeling in several figures is too small and very difficult to read. This is particularly problematic for superindices, such as "Ig1-6", "fe", exponentials of molar concentrations, etc. Also the use of very light (for example yellow) font colors makes it difficult to read. Please,

As per the reviewer's request we have enlarged the labels that have superindices and darkened the lighter color labels to enhance readability.

14) In Fig 2d, it would help to expand the scale of the elution volume son that the peaks do not appear so narrow (there are no visible peaks before 11 ml or after 15 ml). Also the insert plots are redundant (and do not show the values of the axis).

As per the reviewer's request we have expanded the scale of the elution volume and removed the insert plots.

15) The figure legends seem far too short. It would be very helpful to explain the data or structures in more detail.

We have now expanded the figure legends to explain the data and structures in more detail.

REVIEWER COMMENTS

Reviewer #1 (Remarks to the Author):

NCOMMS-21-22916B

Review comment:

Chataigner et al., "Structural insights into the contactin 1 – neurofascin 155 adhesion complex"

Most of my concerns have been properly addressed in the revised manuscript. However, I raise just one suggestion for providing the readers with additional information:

- I understand that SPR signals below ~400 RU are relatively unreliable in the experimental setup in this study. Nevertheless, the quantity of the coupled ligand is a critical factor to evaluate the reliability of SPR experiments. I recommend showing the approximate quantity of the ligand coupled to the sensor chip for each experiment in Supplementary Figure 1a,b.

Reviewer #2 (Remarks to the Author):

The revised manuscript entitled "Structural insights into the contactin 1 – neurofascin 155 adhesion complex" provides missing details and explanations, which very satisfactorily addressed all my comments and concerns. The authors provided all necessary details concerning the challenging refinement of the presented structures and illustrated the manuscript with movies which certainly will help the reader to grasp described structural details.

I must confess, that the decision of authors to provide an anonymous reviewer with both model coordinates in PDB format and reflection files comprising precalculated map coefficients (mtz files) substantially supported the assessment of structure quality. I really appreciate it.

I am convinced that in the present form, the manuscript merits publication.

Reviewer #3 (Remarks to the Author):

The revised manuscript includes new data that strengths some of the conclusions. In particular, the analysis of the homotypic and heterotypic associations of neurofascin 155 and contactin 1 in a cell model system, and the effect of structure-based mutations, adds support to the functional relevance of the contact areas identified from the crystal structures. The inclusion of SEC-MALS data also reinforces the analysis of the homoassociation of neurofascin 155. In general, most of my questions and concerns have been addressed. I still think that there is major conceptual problem with the analysis of the SPR experiments (see next).

Major issue.

1) I still see a major issue with the analysis of the SPR data. The authors say that (i) they used averaged response signals at equilibrium (between 300 and 380 seconds of association phase) and (ii) that only data for neurofascin 155Ig1-6 HM binding to contactin 1 fe CG did not reach equilibrium. Yet, almost none of the sensorgrams in the supplemental figure 1 panels c, d, e, f, g, h, I, and j reached a plateau during the binding or association phase (at least none of the sensorgrams of the 6 higher analyte concentrations of each set). That is, those curves did not reach equilibrium. By definition, analysis of binding isotherms, such as those shown in supplemental figure 1 a, b, and k, and in figure 1b, requires that each data point is measured at equilibrium. Sorry, but it is a simple as this: data obtained at non-equilibrium conditions cannot be used to analyze equilibrium binding curves. Unfortunately, saying that this Kd values are approximations does not solve the problem. Therefore,

reporting the $K_d=0.6 \mu\text{M}$ for the heteroassociation determined this way is not adequate. In contrast, the new data in supplemental fig 1I has been measured to equilibrium, which allows analyzing the binding curve as shown in supplemental figure 1k. The SPR data seems to have been measured in conditions (low ligand densities) that may allow analyzing and determining the kinetic parameters of the interaction. I agree with referee #1 that the lack of plateau in the sensorgrams, in particular at high concentrations of analyte, might reflect a component of unspecific interaction. The new data in supplemental fig 1I does not exclude this possibility. If possible, the authors should do kinetic analyses of the SPR sensorgrams, which should allow the determination of association and dissociation rate constants from which it should be possible to estimate the K_d .

Minor issues

2) The new section of the results describing cell-clustering analysis (lines 370-414) is written in present tense. Please, use past tense.

3) In table 1, the high resolution shell of the data collection of NF155Ig1-6 was previously reported as 3.02-2.97 Å. Now it is reported as 3.0-2.9 in table 1. Please, correct. Rounding up 2.97 to 2.9 is incorrect. If the authors need to use two decimals, please do so.

4) Supplemental Figure 5: please include the MW estimated from the analysis of the sedimentation velocity data. That will add valuable details to the "monomer", "dimer", "trimer" classification, which might be an oversimplification.

5) Several new molecular movies are included to address issues with the representations of the structures. As a comment, it is a pity that structural elements such as domains are not labeled or identified in those movies. Hence, the usefulness of these movies is limited.

6) References are not formatted consistently. At this advanced level, and with all due respect, this is not serious and reflects poor care in preparing the manuscript.

Reviewer #4 (Remarks to the Author):

The data from the "surface plasmon resonance" experiments do not have the required quality for the estimation of K_D -values or a qualitative comparative analysis of the interactions studied. The results presented consequently lack experimental support.

Firstly, the method is described as "Surface plasmon resonance (SPR)". But SPR is a very broad term that is misleading in this context as it here refers to a specific imaging technology with arrays of spotted protein. It consequently differs considerably from conventional SPR biosensor technologies in how experiments are performed and the information that can reliably be obtained. The key principle for the method is not simply captured by the term SPR although the readout is the same as other SPR-based methods. It must be clarified to the reader that the method is "SPR imaging using spotted protein arrays". (It would be just as misleading to use the term "fluorescence" to describe any technology that uses fluorescence as the readout without describing the actual method.)

Secondly, the study of protein-protein interactions is challenging. Imaging SPR is expected to have the same challenges as conventional SPR biosensor methods and is not suited for determining the kinetics or affinities for such interactions.

There are several problems with the presented data. Specifically:

- The sensorgrams shown in Supplementary Figure 1 c-j exhibit several complexities which prevents

global fitting of the sensorgrams using kinetic models.

- o There is a large shift in the signal upon injection of the sample that does not represent a 1:1 interaction in one reversible step. It could be a mismatch between sample and running buffers, but more likely represents two interaction events, one primary interaction seen at low concentrations, followed by multiple secondary interactions with lower affinity at higher concentrations or of a larger complex interacting at higher concentrations (dimers or higher order complexes).
- o There is a drift in the signal and steady state is not reached within the time for the injection of analyte. It is most noticeable for the apparent secondary interactions, but might also occur for the primary interaction.
- o There is no saturation of the signal at high concentrations suggesting that the apparent secondary interactions are non-specific aggregations rather than mechanistically well-defined interactions.
- A steady-state analysis based on average signal taken from the association phase is consequently also meaningless although the data in Supplementary Figure 1 a and k looks as though it could represent the first part of a sigmoid curve (log concentration). The fitting using a Langmuir model (for a reversible 1-step 1:1 interaction) requires "clean data" without any underlying complexities. Without a B_{max} , the estimates from this steady-state analysis are also meaningless. The report points used for the analysis do not represent steady-state data the type required for such an analysis. The KD -values can therefore not be seen as meaningful approximations.
- In principle it would be possible to do a qualitative, comparative, analysis using the data that is not influenced by the complexities seen. In this case it would include only the low concentration sensorgrams that reach an apparent steady state.

The dimerization of neurofascin may be a complicating factor for these experiments.

Detailed comments

Line 125-125: This statement is incorrect "We determined direct interaction of contactin 1Ig1-6 with neurofascin 155Ig1-6 with a K_d of $0.6 \mu M$ ". It is not possible to establish any KD -values from the data generated.

Line 181-184: There is no experimental support for these statements "We do not observe a strong difference in neurofascin 155Ig1-6 interaction affinity to contactin 1 immunoglobulin and full ectodomain (fe) segments containing either mannose rich or complex glycans (Fig. 1b, Supplementary Fig. 1). On the other hand, interaction of contactin 1 to neurofascin 155Ig1-6 containing mannose rich glycans is 2.4-3.2 fold stronger compared to neurofascin 155Ig1-6 containing complex glycans. Possibly the glycans on Ig5 of neurofascin 155, of which Asn494 is conserved, and that interface with the glycan on Asn258 of contactin 1, play a role in modulating the affinity (Fig. 1 b,d)." It is not possible to establish any KD -values from the data generated and comparisons between the "steady-state" curves are not meaningful.

Line 244: The statement "substantially weaker than the contactin 1 – neurofascin 155 affinity of $0.6 \mu M$ (Fig. 1b)" lacks experimental support.

Line 452: The statement "The contactin 1 – neurofascin 155 complex has a higher affinity ($0.6 \mu M$) compared to neurofascin 155 dimerization ($5-30 \mu M$), suggesting a preference for heterophilic complex formation over homodimerization." lacks experimental support.

Line 474: The statement "In SPR experiments we show that contactin 1 – neurofascin 155 interactions still occur when both proteins have the same glycan type, suggesting that glycan microheterogeneity, steric properties of the full-length molecules, or structural constraints of the cellular context may additionally modulate transcellular interactions." lacks experimental support.

Line 639: The method described for analysing data is not appropriate for the current data set due to the complexities of the system. "Response units based on averaged response signal at equilibrium, i.e. between 300 and 380 seconds of association phase were plotted against the analyte concentration and modeled with a 1:1 Langmuir binding model to calculate the KD and the maximum analyte binding

(Bmax).

Line 642: It is incorrect to state that KDs are approximations, the values estimated are simply not equilibrium constants at all. "In experiments where equilibrium is not reached, for example for neurofascin 155Ig1-6 HM binding to contactin 1fe CG, the KD's are approximations."

Figure 1b: It is not clear from the graph that it is neurofascin that is the analyte at different concentrations (the orange triangles are stated to represent contactin). Supplemental figure 1 a and b have the same problem.

Supplementary Figure 1: the figure legend should be adjusted with respect to the comments above (the type of SPR technique, experimental setup and data analysis etc.).

REVIEWER COMMENTS TO NCOMMS-21-22916B

We would like to thank the reviewers for their comments and suggestions. We have added new SPR data in which equilibrium is reached at each concentration and reanalysed the SPR data where we compare different glycoforms in a qualitative, comparative manner. Below we detail our response to each comment.

Reviewer #1:

Review comment:

Chataigner et al., “Structural insights into the contactin 1 – neurofascin 155 adhesion complex”

Most of my concerns have been properly addressed in the revised manuscript. However, I raise just one suggestion for providing the readers with additional information:

- I understand that SPR signals below ~400 RU are relatively unreliable in the experimental setup in this study. Nevertheless, the quantity of the coupled ligand is a critical factor to evaluate the reliability of SPR experiments. I recommend showing the approximate quantity of the ligand coupled to the sensor chip for each experiment in Supplementary Figure 1a,b.

We apologise to the reviewer as there may have been unclarity in our previous reply. We did not intend to state that the SPR signal is unreliable below ~400 RU, it is very reliable, however the reviewer is correct that the determination of the amount of deposited ligand on the sensor surface is relatively unreliable below 400 RU due to the offline coupling of the ligand. The determination of the Bmax from the model on the other hand is very reliable even at very low ligand densities well below 100 RU.

On suggestion of reviewer #3 and 4 we have updated the SPR results. We have taken out our data where equilibrium is not reached and have replaced part of it by new data at which equilibrium is obtained at each concentration (figure 1b and supplementary figure 1a-d). This shows that contactin 1^{Ig1-6} and contactin 1^{fe} bind to neurofascin 155^{Ig1-6} with similar K_D's of 0.2 and 0.3 μM, respectively. In addition, on suggestion of reviewer #4 we have reanalysed part of the data in a qualitative manner where we compare different glycan versions of the proteins (supplementary figure 1e-g). This new analysis is in line with our previous observations. We have reworded the text to reflect the new data. We report the quantity of ligand coupled to the sensor chip together with the calculated theoretical Bmax for those experiments in which we used a quantitative analysis (supplementary figure 1a-d).

Reviewer #2 (Remarks to the Author):

The revised manuscript entitled “Structural insights into the contactin 1 – neurofascin 155 adhesion complex” provides missing details and explanations, which very satisfactorily addressed all my comments and concerns. The authors provided all necessary details concerning the challenging refinement of the presented structures and illustrated the manuscript with movies which certainly will help the reader to grasp described structural

details.

I must confess, that the decision of authors to provide an anonymous reviewer with both model coordinates in PDB format and reflection files comprising precalculated map coefficients (mtz files) substantially supported the assessment of structure quality. I really appreciate it.

I am convinced that in the present form, the manuscript merits publication.

We thank the reviewer for the kind words.

Reviewer #3 (Remarks to the Author):

The revised manuscript includes new data that strengthens some of the conclusions. In particular, the analysis of the homotypic and heterotypic associations of neurofascin 155 and contactin 1 in a cell model system, and the effect of structure-based mutations, adds support to the functional relevance of the contact areas identified from the crystal structures. The inclusion of SEC-MALS data also reinforces the analysis of the homoassociation of neurofascin 155. In general, most of my questions and concerns have been addressed. I still think that there is major conceptual problem with the analysis of the SPR experiments (see next).

Thank you for the kind words and the possibility to address the remaining comments, see below for more details.

Major issue.

1) I still see a major issue with the analysis of the SPR data. The authors say that (i) they used averaged response signals at equilibrium (between 300 and 380 seconds of association phase) and (ii) that only data for neurofascin 155Ig1-6 HM binding to contactin 1 fe CG did not reach equilibrium. Yet, almost none of the sensorgrams in the supplemental figure 1 panels c, d, e, f, g, h, I, and j reached a plateau during the binding or association phase (at least none of the sensorgrams of the 6 higher analyte concentrations of each set). That is, those curves did not reach equilibrium. By definition, analysis of binding isotherms, such as those shown in supplemental figure 1 a, b, and k, and in figure 1b, requires that each data point is measured at equilibrium. Sorry, but it is as simple as this: data obtained at non-equilibrium conditions cannot be used to analyze equilibrium binding curves. Unfortunately, saying that this Kd values are approximations does not solve the problem.

Therefore, reporting the Kd=0.6 uM for the heteroassociation determined this way is not adequate.

In contrast, the new data in supplemental fig 1I has been measured to equilibrium, which allows analyzing the binding curve as shown in supplemental figure 1k.

The SPR data seems to have been measured in conditions (low ligand densities) that may allow analyzing and determining the kinetic parameters of the interaction. I agree with referee #1 that the lack of plateau in the sensorgrams, in particular at high concentrations of analyte, might reflect a component of unspecific interaction. The new data in supplemental fig

It does not exclude this possibility. If possible, the authors should do kinetic analyses of the SPR sensorgrams, which should allow the determination of association and dissociation rate constants from which it should be possible to estimate the K_d .

We thank the reviewer for commenting on the absence of equilibrium in most of the SPR data and pointing out that equilibrium is adequately reached as shown in supplementary figure 1k. As suggested by this reviewer and reviewer #4 we have replaced the SPR data in supplementary fig. 1 panels c-j by new data in which equilibrium is reached at each concentration and by qualitative reanalysis of part of the data (on suggestion of reviewer #4).

The new data obtained from equilibrium states, are fitted to a 1:1 Langmuir model (Fig. 1 b and Supplementary fig 1 a-d). The calculated B_{max} , derived from the Langmuir model, is 30-50% of the theoretical B_{max} calculated from the deposited ligand (both B_{max} values are reported in supplementary figure 1 a-d). The combination of the equilibrium at each concentration, the saturation at high analyte concentration and the ratio of calculated B_{max} versus theoretical B_{max} suggest that the interaction is specific. Also, our cell clustering assays, albeit in a very different setting, indicate that the interaction between contactin 1 and neurofascin 155 is specific.

On suggestion of reviewer #4 we have performed a qualitative, comparative analysis of the different glycan states at low analyte concentrations (supplementary figure 1e-g). In line with our previous conclusions, we show that there is better binding of neurofascin 155 HM to different contactin 1 ligands compared to neurofascin 155 CG. Please note that the neurofascin 155 versions are only compared when probed on the same contactin 1 ligand region on the sensor surface, guaranteeing comparison at the same amount of deposited ligand, and that neurofascin 155 CG was probed first followed by neurofascin 155 HM. We have adjusted the manuscript to reflect this qualitative, comparative analysis.

Minor issues

2) The new section of the results describing cell-clustering analysis (lines 370-414) is written in present tense. Please, use past tense.

We have now written this section in the past tense.

3) In table 1, the high resolution shell of the data collection of NF155Ig1-6 was previously reported as 3.02-2.97 Å. Now it is reported as 3.0-2.9 in table 1. Please, correct. Rounding up 2.97 to 2.9 is incorrect. If the authors need to use two decimals, please do so.

We thank the reviewer for spotting this. The error in rounding up has now been corrected in table 1. Please note that we rounded up the resolution shells in the previous round of revisions based on a suggestion from reviewer 2. We decided to not round up the 3.02-2.97 part as that would round up to 3.0-3.0 which could be confused with an empty resolution shell.

4) Supplemental Figure 5: please include the MW estimated from the analysis of the sedimentation velocity data. That will add valuable details to the "monomer", "dimer", "trimer" classification, which might be an oversimplification.

We have added the MW estimates and the best-fit frictional ratio from the sedimentation velocity data to each panel in supplementary figure 5.

5) Several new molecular movies are included to address issues with the representations of the structures. As a comment, it is a pity that structural elements such as domains are not labeled or identified in those movies. Hence, the usefulness of these movies is limited.

We have now added domain labels to the first part of each movie to point out the domains to the reader. We thank the reviewer for the suggestion.

6) References are not formatted consistently. At this advanced level, and with all due respect, this is not serious and reflects poor care in preparing the manuscript.

We have reformatted the references for consistency.

Reviewer #4 (Remarks to the Author):

The data from the “surface plasmon resonance” experiments do not have the required quality for the estimation of KD-values or a qualitative comparative analysis of the interactions studied. The results presented consequently lack experimental support.

Firstly, the method is described as “Surface plasmon resonance (SPR)”. But SPR is a very broad term that is misleading in this context as it here refers to a specific imaging technology with arrays of spotted protein. It consequently differs considerably from conventional SPR biosensor technologies in how experiments are performed and the information that can reliably be obtained. The key principle for the method is not simply captured by the term SPR although the readout is the same as other SPR-based methods. It must be clarified to the reader that the method is “SPR imaging using spotted protein arrays”. (It would be just as misleading to use the term “fluorescence” to describe any technology that uses fluorescence as the readout without describing the actual method.)

As suggested by the reviewer we have now elaborated in the methods section on the SPR imaging, as in: “Continuous flow microspotting was used to deposit an array of c-terminally biotinylated proteins” and “Surface plasmon resonance imaging experiments were performed on a MX96 SPRi instrument (IBIS Technologies)”. We have also updated the legend of supplementary fig. 1 as in: “Surface plasmon resonance imaging interaction data using spotted protein arrays of ...”

Secondly, the study of protein-protein interactions is challenging. Imaging SPR is expected to have the same challenges as conventional SPR biosensor methods and is not suited for determining the kinetics or affinities for such interactions.

There are several problems with the presented data. Specifically:

- The sensorgrams shown in Supplementary Figure 1 c-j exhibit several complexities which prevents global fitting of the sensorgrams using kinetic models.*

o There is a large shift in the signal upon injection of the sample that does not represent a 1:1 interaction in one reversible step. It could be a mismatch between sample and running buffers, but more likely represents two interaction events, one primary interaction seen at low concentrations, followed by multiple secondary interactions with lower affinity at higher concentrations or of a larger complex interacting at higher concentrations (dimers or higher order complexes).

o There is a drift in the signal and steady state is not reached within the time for the injection of analyte. It is most noticeable for the apparent secondary interactions, but might also occur for the primary interaction.

o There is no saturation of the signal at high concentrations suggesting that the apparent secondary interactions are non-specific aggregations rather than mechanistically well-defined interactions.

• A steady-state analysis based on average signal taken from the association phase is consequently also meaningless although the data in Supplementary Figure 1 a and k looks as though it could represent the first part of a sigmoid curve (log concentration). The fitting using a Langmuir model (for a reversible 1-step 1:1 interaction) requires “clean data” without any underlying complexities. Without a Bmax, the estimates from this steady-state analysis are also meaningless. The report points used for the analysis do not represent steady-state data the type required for such an analysis. The KD-values can therefore not be seen as meaningful approximations.

• In principle it would be possible to do a qualitative, comparative, analysis using the data that is not influenced by the complexities seen. In this case it would include only the low concentration sensorgrams that reach an apparent steady state.

The dimerization of neurofascin may be a complicating factor for these experiments.

As suggested by this reviewer and reviewer #3, we have now replaced our SPR data as reported in supplementary figure 1 c-j and the analysis as reported in figure 1b and supplementary figure 1a,b, (i.e all the data in which equilibrium is not reached) with new SPR experiments in which each injection reaches equilibrium (new figure 1b and supplementary figure 1a-d) and with a qualitative, comparative, analysis using only low concentration analyte data (new supplementary figure 1e-g).

The new data, in which steady-state is reached for each injection and saturation is obtained at high analyte concentration, is fitted by a 1:1 Langmuir model (figure 1b and supplementary figure, 1a-d) indicating a K_D of 0.2 and 0.3 μM for the interaction of contactin 1 (Ig1-6 or full ectodomain) with neurofascin 155 Ig1-6. This supports our main conclusion that our purified contactin 1 and neurofascin 155 interact. The calculated Bmax is 30-50% of the theoretical Bmax as calculated from the deposited ligand (both are reported in supplementary figure 1a-d). This may indicate that not all analyte binding sites on the ligand coupled to the surface (i.e. contactin 1^{Ig1-6}) are available for analyte binding, similar to the 50% exposed binding sites in the contactin 1 zipper structure that we report on (we have not looked into this any further).

We would like to thank the reviewer for the suggestion to replace our quantitative SPR analysis of the different glycoforms with a qualitative, comparative analysis using low analyte concentrations. We now focus on the 160 nM injection of neurofascin 155 HM and CG and only compared them when probed on the same contactin 1 ligand sensor region (ensuring the same amount of deposited ligand) (supplementary figure 1e-g). In this experiment neurofascin

155 CG was first injected followed by neurofascin 155 HM. The new qualitative analysis supports our previous observation that binding of neurofascin 155 containing high mannose glycans to contactin 1 is slightly better compared to neurofascin 155 containing complex glycans. We have not included the data of the contactin 1^{Ig1-6} CG ligand because the 160 nM analyte data was noisy for this ligand region, most likely due to low levels of ligand deposition. The contactin 1^{fe} CG ligand data makes up for this omission (supplementary figure 1g) and supports our previous conclusions.

The new SPR data and the qualitative analysis of the different glycoforms at low analyte concentrations support our earlier conclusions. We have rephrased the relevant sections in the manuscript and describe these below in our response to the detailed comments from the reviewer.

Detailed comments

Line 125-125: This statement is incorrect “We determined direct interaction of contactin 1Ig1-6 with neurofascin 155Ig1-6 with a Kd of 0.6 μM”. It is not possible to establish any KD-values from the data generated.

We have added three new SPR experiments (figure 1b and supplementary figure 1a-d), together with the already reported experiment in supplementary figure 1L in the old version, in which steady-state is reached for each injection and saturation is obtained at high analyte concentration. The Kd for the contactin 1^{Ig1-6} with neurofascin 155^{Ig1-6} interaction is 0.22 μM. We have changed the value accordingly in the text.

Line 181-184: There is no experimental support for these statements “We do not observe a strong difference in neurofascin 155Ig1-6 interaction affinity to contactin 1 immunoglobulin and full ectodomain (fe) segments containing either mannose rich or complex glycans (Fig. 1b, Supplementary Fig. 1). On the other hand, interaction of contactin 1 to neurofascin 155Ig1-6 containing mannose rich glycans is 2.4-3.2 fold stronger compared to neurofascin 155Ig1-6 containing complex glycans. Possibly the glycans on Ig5 of neurofascin 155, of which Asn494 is conserved, and that interface with the glycan on Asn258 of contactin 1, play a role in modulating the affinity (Fig. 1 b,d).” It is not possible to establish any KD-values from the data generated and comparisons between the “steady-state” curves are not meaningful.

We have rephrased this section to: “We do not observe a strong difference in neurofascin 155^{Ig1-6} interaction affinity to contactin 1 immunoglobulin or full ectodomain (fe) segments containing mannose rich glycans. Interaction of contactin 1 to neurofascin 155^{Ig1-6} containing mannose rich glycans is stronger compared to neurofascin 155^{Ig1-6} containing complex glycans although we have not quantified the difference in binding affinity”.

Line 244: The statement “substantially weaker than the contactin 1 – neurofascin 155 affinity of 0.6 μM (Fig. 1b)” lacks experimental support.

We now refer here to the new SPR data in which steady-state is reached for each injection and saturation is reached at high analyte concentration.

Line 452: The statement “The contactin 1 – neurofascin 155 complex has a higher affinity (0.6 μ M) compared to neurofascin 155 dimerization (5-30 μ M), suggesting a preference for heterophilic complex formation over homodimerization.” lacks experimental support.

We now refer here to the new quantitative SPR data.

Line 474: The statement “In SPR experiments we show that contactin 1 – neurofascin 155 interactions still occur when both proteins have the same glycan type, suggesting that glycan microheterogeneity, steric properties of the full-length molecules, or structural constraints of the cellular context may additionally modulate transcellular interactions.” lacks experimental support.

We have slightly rephrased this and refer to the new data: “In SPR experiments we show that contactin 1 – neurofascin 155 interactions still occur when both proteins have the same high-mannose glycan type (Fig. 1b), suggesting that glycan microheterogeneity, steric properties of the full-length molecules, or structural constraints of the cellular context may additionally modulate transcellular interactions”.

Line 639: The method described for analysing data is not appropriate for the current data set due to the complexities of the system. “Response units based on averaged response signal at equilibrium, i.e. between 300 and 380 seconds of association phase were plotted against the analyte concentration and modeled with a 1:1 Langmuir binding model to calculate the KD and the maximum analyte binding (Bmax).

We have added new SPR data in which steady-state is reached for each injection and saturation is reached at high analyte concentration. These steady-state data points are fitted by a Langmuir model representing a reversible 1-step 1:1 interaction (supplementary figure 1a-d).

Line 642: It is incorrect to state that KDs are approximations, the values estimated are simply not equilibrium constants at all. “In experiments where equilibrium is not reached, for example for neurofascin 155Ig1-6 HM binding to contactin 1fe CG, the KD’s are approximations.”

As suggested by the reviewer we have replaced our SPR data where equilibrium is not reached by new data and reanalysed only the data where equilibrium is obtained. We have replaced the above sentence by: “Interaction experiments shown in Supplementary Fig. 1e-g were not quantified”.

Figure 1b: It is not clear from the graph that it is neurofascin that is the analyte at different concentrations (the orange triangles are stated to represent contactin). Supplemental figure 1a and b have the same problem.

We have adapted the figures (figure 1b and supplementary figure 1a-d) to make it clear that neurofascin 155 is the analyte by labeling the x-axis by the analyte, and present the remaining information in clear inset legends.

Supplementary Figure 1: the figure legend should be adjusted with respect to the comments above (the type of SPR technique, experimental setup and data analysis etc.).

The legend of supplementary figure 1 was changed to: “Contactin 1 and neurofascin 155 interact. **a-d** Surface plasmon resonance imaging interaction data using spotted protein arrays of contactin 1^{fc} HM and contactin 1^{Ig1-6} HM ligands with neurofascin 155^{Ig1-6} HM analyte. a-b and c-d are technical duplicates, i.e independent positions on the sensor surface. SPR sensorgrams, with the association, dissociation and regeneration phases indicated are shown left. Equilibrium binding data versus analyte concentration modeled with a 1:1 Langmuir binding model are shown right. The theoretical Bmax is calculated as the amount of deposited ligand on the surface corrected for the difference in molecular weight between the ligand and the analyte and assuming a 1:1 interaction. **e-g** Qualitative comparison of neurofascin 155^{Ig1-6} HM (black lines) and CG (green lines) versions binding to contactin 1^{fc} HM (**e**) and CG (**g**), and contactin 1^{Ig1-6} HM (**f**) ligands. The lower response of neurofascin 155^{Ig1-6} CG compared to neurofascin 155^{Ig1-6} HM at equivalent concentrations indicates the CG version of neurofascin 155^{Ig1-6} binds with lower affinity to contactin 1 compared to the neurofascin 155^{Ig1-6} HM version”.

REVIEWERS' COMMENTS

Reviewer #4 (Remarks to the Author):

The authors have adequately revised the manuscript to better reflect the exact experimental approach (methods section) and presented improved figures and legends that allow the reader to better understand the experiments. The interpretation of the results is now based on quantitative data when possible, but otherwise using a qualitative approach that allows reasonable conclusions to be drawn.

Minor points:

1. Change heading in methods section from "Surface plasmon resonance" to "Surface plasmon resonance imaging".
2. The abbreviations used in the figures for the ligands and analyses are not clearly explained in the text of figure legends. It is difficult to simply look at the figure and understand the construct or form of protein studied in each experiment. This is particularly difficult due to the small size of the figures and the text inside the figures (Figure 1b) and supplementary figures.

REVIEWER COMMENTS TO NCOMMS-21-22916C

Reviewer #4:

The authors have adequately revised the manuscript to better reflect the exact experimental approach (methods section) and presented improved figures and legends that allow the reader to better understand the experiments. The interpretation of the results is now based on quantitative data when possible, but otherwise using a qualitative approach that allows reasonable conclusions to be drawn.

Minor points:

1. Change heading in methods section from "Surface plasmon resonance" to "Surface plasmon resonance imaging".

We have changed the heading as requested.

2. The abbreviations used in the figures for the ligands and analyses are not clearly explained in the text of figure legends. It is difficult to simply look at the figure and understand the construct or form of protein studied in each experiment. This is particularly difficult due to the small size of the figures and the text inside the figures (Figure 1b) and supplementary figures.

We have now explained the abbreviations in each of the figures and supplementary figures.